# Ubiquitination of RIPK1 suppresses programmed cell death by regulating RIPK1 kinase activation during embryogenesis

Xixi Zhang[1,5], Haiwei Zhang[1,5], Chengxian Xu[1], Xiaoming Li[1], Ming Li[1], Xiaoxia Wu[1], Wenjuan Pu [2], Bin Zhou [2], Haikun Wang[3], Dali Li[4], Qiurong Ding [1], Hao Ying[1], Hui Wang[1] & Haibing Zhang [1]

The ubiquitination status of RIPK1 is considered to be critical for cell fate determination. However, the in vivo role for RIPK1 ubiquitination remains undefined. Here we show that mice expressing RIPK1[K376R] which is defective in RIPK1 ubiquitination die during embryogenesis. This lethality is fully rescued by concomitant deletion of *Fadd* and *Ripk3* or *Mlkl*. Mechanistically, cells expressing RIPK1[K376R] are more susceptible to TNF-α induced apoptosis and necroptosis with more complex II formation and increased RIPK1 activation, which is consistent with the observation that *Ripk1[K376R/K376R]* lethality is effectively prevented by treatment of RIPK1 kinase inhibitor and is rescued by deletion of Tnfr1. However, *Tnfr1[−/−] Ripk1[K376R/K376R]* mice display systemic inflammation and die within 2 weeks. Significantly, this lethal inflammation is rescued by deletion of *Ripk3*. Taken together, these findings reveal a critical role of Lys376-mediated ubiquitination of RIPK1 in suppressing RIPK1 kinase activity–dependent lethal pathways during embryogenesis and RIPK3-dependent inflammation postnatally.

[1] CAS Key Laboratory of Nutrition, Metabolism and Food Safety, Shanghai Institute of Nutrition and Health, Shanghai Institutes for Biological Sciences, University of Chinese Academy of Sciences, Chinese Academy of Sciences, 100864 Shanghai, China. [2] The State Key Laboratory of Cell Biology, CAS Center for Excellence on Molecular Cell Science, Shanghai Institute of Biochemistry and Cell Biology, University of Chinese Academy of Sciences, Chinese Academy of Sciences, 100864 Shanghai, China. [3] Key Laboratory of Molecular Virology and Immunology, Unit of the Regulation of Immune Cell Differentiation, Institute Pasteur of Shanghai, Shanghai Institutes for Biological Sciences, Chinese Academy of Sciences, 100864 Shanghai, China. [4] Shanghai Key Laboratory of Regulatory Biology, Institute of Biomedical Sciences and School of Life Sciences, East China Normal University (ECNU), 200062 Shanghai, China. [5] These authors contributed equally: Xixi Zhang, Haiwei Zhang. Correspondence and requests for materials should be addressed to H.Z. (email: hbzhang@sibs.ac.cn)

RIPK1, an essential signaling node in various innate immune signaling pathways, is most extensively studied in the TNFR1 signaling pathway. Ligation of TNFR1 results in the rapid formation of a signaling complex termed complex I or TNFR1-Siganaling complex (TNFR1-SC) which is composed of TRADD, RIPK1, TRAF2, and cIAP1/2[1,2]. Following recruitment, RIPK1 is polyubiquitinated by multiple forms of ubiquitin chains which provides docking sites for two protein complexes, the TAB/TAK complex comprising TAK1 and TAB2/3, as well as the IKK complex comprising NEMO, IKKα, and IKKβ[3–6]. The individual formation of these two complexes leads to activation of NF-κB and MAPK signaling pathways and therefore promotes cell survival. Nevertheless, under certain circumstances when the pro-survival signal emanating from complex I is perturbed, RIPK1 dissociates from complex I and recruits FADD and Caspase-8 to form complex IIa, and this leads to initiation of apoptosis[1,7–11]. However, when apoptosis is inhibited, RIPK1 can interact with RIPK3 to form an amyloid-like complex termed complex IIb or necrosome, resulting in activation of necroptosis[12–17].

Recently, by extensive genetic studies on RIPK1 signaling pathways, the in vivo role of RIPK1 in regulating both cell survival and cell death has been revealed. Genetic deletion of *Ripk1* in animals leads to postnatal lethality with widespread cell death in lymphoid and adipose lineages[18]. Ablation of *Caspase-8/Fadd* and *Ripk3* allows for normal development and maturation of Ripk1-deficient mice[19–22]. Similarly, conditional deletion of Ripk1 in intestinal epithelial cells (IECs) results in premature death in mice accompanied by extensive apoptosis in intestine and ensuing inflammation[23,24]. These phenotypes are largely resolved in mice lacking intestinal *Fadd/Caspase-8* or both *Fadd* and *Ripk3*[23,24]. In addition, the mice with keratinocyte-specific *Ripk1* deficiency progressively develop severe inflammatory skin lesions that are fully prevented by deletion of *Ripk3* or *Mlkl*[24]. Therefore, RIPK1 seems to be a suppressor of lethal pathways engaged by FADD/Caspase-8 and RIPK3/MLKL. However, RIPK1 is also suggested to mediate cell death in some contexts. Ablation of *Ripk1* prevents early embryonic lethality induced by *Caspase-8* or *Fadd*, although these mice succumb to early postnatal lethality similar to *Ripk1* deficient mice[21,22,25]. Another striking study showed that mice with homozygous *Ripk3^{D161N}* died at E10.5 but were completely rescued by co-deletion of *Ripk1*[26]. These investigations revealed an opposite function of RIPK1 which can also induce cell death. Therefore, the paradoxical roles of RIPK1 raise the following fundamental question: what is the switch for two opposing functions of RIPK1?

RIPK1 consists of an N-terminal serine/threonine kinase domain, an intermediate domain, and a C-terminal death domain (DD)[27,28]. The kinase activity is essential for induction of necroptosis and RIPK1-dependent apoptosis but is dispensable for NF-κB activation[29–31]. Animals with kinase-inactive mutant RIPK1 develop normally and are protected from TNF-α induced necroptosis in vitro and in vivo, suggesting that the pro-survival function of RIPK1 does not require RIPK1 kinase activity[32,33]. The intermediate domain contains a RIP homotypic interaction motif (RHIM) which binds the RHIM of RIPK3 to initiate necroptosis, whereas the genetic studies reveal a critical role of RHIM in inhibiting ZBP1/RIPK3/MLKL-dependent necroptosis during development and inflammation[34–36]. In addition, a recent work has shown that DD-mediated RIPK1 dimerization is crucial for RIPK1 activation, and mice with mutant DD are resistant to RIPK1-dependent apoptosis and necroptosis[37]. Thus, although these genetic studies have demonstrated that RIPK1 is involved in mediating apoptosis, necroptosis and cell survival, the mechanism by which RIPK1 regulates these pathways in vivo remains unclear.

Given that the ubiquitination status of RIPK1 is critical for the transition of complex I to complex II, a clue to the answer may come from a major functional ubiquitination site K377 of human RIPK1, which provides a docking site for the K-63 linked polyubiquitin chain and mediates the recruitment of TAK1 and IKK complexes[3,5,38,39]. In vitro studies have shown that substitution of Lys-377 with arginine (K377R) blocked RIPK1 ubiquitination and TNF-α-mediated NF-κB activation and sensitized cells to TNF-α-induced apoptosis[3,5,38,39]. However, the in vivo function of K377, which is the key to uncover the enigma of the paradoxical roles of RIPK1, has not been established.

In this study, we generate mice endogenously expressing RIPK1^{K376R}, a single amino acid change in K376 of mouse RIPK1, which is homologous to K377 of human RIPK1. Strikingly, mice bearing *Ripk1^{K376R/K376R}* die at embryonic day 12.5 (E12.5) with excessive cell death in embryonic tissues and the yolk sac. Accordingly, Mouse embryonic fibroblasts (MEFs) expressing RIPK1^{K376R} are defective in TNF-α-induced ubiquitination and are more sensitive to TNF-α-induced apoptosis and necroptosis. The excessive cell death in mutant embryos which can be effectively prevented by Nec-1 treatment is proved to be dependent on the kinase activity of RIPK1. Intriguingly, *Ripk1^{K376R/−}* mice with only half amounts of mutant RIPK1^{K376R} are viable although these mice develop systemic inflammation after birth. Besides, ablation of *Fadd* and *Ripk3/Mlkl* rescues *Ripk1^{K376R/K376R}* mice from embryonic lethality and allows the animals to grow into fertile adults, indicating that the lethal phenotypes of mutant mice are caused by FADD-dependent apoptosis and RIPK3/MLKL dependent necroptosis. Furthermore, deletion of *Tnfr1* rescues *Ripk1^{K376R/K376R}* mice at the embryonic stage but fails to prevent the postnatal systemic inflammation of the mutant mice. Importantly, *Ripk3* deficiency prevents lethal inflammation of *Tnfr1^{−/−} Ripk1^{K376R/K376R}* mice, suggesting that ubiquitination of RIPK1 is also involved in regulating inflammation during postnatal development. Thus, our findings provide genetic evidences that Lys376-mediated ubiquitination of RIPK1 plays critical roles in regulating both embryogenesis and inflammation processes.

## Results

***Ripk1^{K376R/K376R}* mice die during embryogenesis**. To address the potential role of RIPK1 ubiquitination in vivo, we generated knock-in mice with Lysine on a key ubiquitination site mutated to Arginine (K376R) (Fig. 1a). Unexpectedly, unlike *Ripk1^{−/−}* mice that died within 3 days after birth, *Ripk1^{K376R/K376R}* mice died during embryogenesis as intercrossing of heterozygous mice only generated heterozygous and wild-type (WT) offspring (Fig. 1b). *Ripk1^{K376R/+}* mice had the same normal life span as WT littermates, excluding the possibility that RIPK1K376R acted as a dominant negative mutant. To gain more insight into the lethality of *Ripk1^{K376R/K376R}* mice, we performed timed pregnancies by mating heterozygous animals. The results showed that *Ripk1^{K376R/K376R}* embryos and their yolk sacs appeared normal at E11.5 (Fig. 1c). However, staining for TUNEL revealed increasing dead cells in fetal livers of the mutant embryos (Fig. 1d). At E12.5, although the appearances of *Ripk1^{K376R/K376R}* embryos were normal, histological examination showed remarkable tissue losses in parts of fetal livers (Fig. 1c, d). Immunoblot analysis showed activated caspase-3 and the cleavage of PARP, as well as aggregations of RIPK1 and RIPK3 were clearly detected in body tissues of mutant embryos, suggesting that activation of apoptosis and necroptosis contributes to the cell death in mutant embryos (Fig. 1f). Besides, immunostaining of yolk sacs for VE-cadherin revealed obvious vascular abnormalities with remarkably enhanced caspase-3 activation in the yolk sacs of mutant

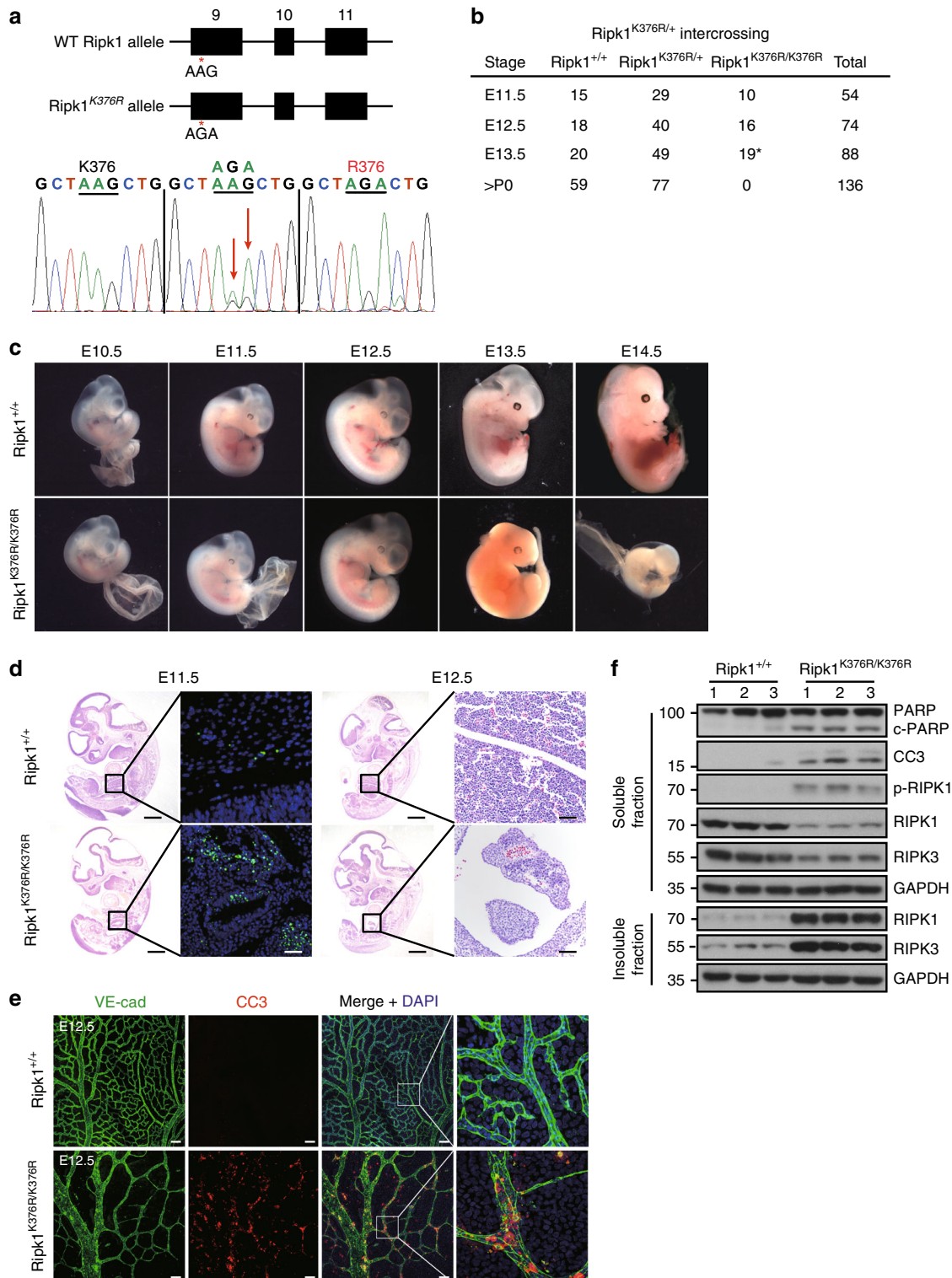

**Fig. 1** *Ripk1K376R/K376R* mice die during embryogenesis. **a** Organization of the *Ripk1K376R* mutant allele. Lysine (AAG) was mutated to Arginine (AGA) at the 376 position in RIPK1. The mutation was confirmed by sequencing. **b** Observed numbers of embryos or live born pups of the indicated genotypes at various developmental stages from *Ripk1K376R/+* mice intercrosses. Asterisk indicates the abnormal embryos. **c** Whole-mount dark field images of embryos with indicated genotypes. Images are representative of embryos from E10.5 (n = 5), E11.5 (n = 10), E12.5 (n = 16), E13.5 (n = 19), E14.5 (n = 5). **d** Representative Hematoxylin and eosin (H&E)-stained and TUNEL-stained fetal livers from mouse embryos of the indicated genotypes (Scale bars, 1 mm). Images are representative of fetal livers from wild-type (n = 3) and *Ripk1K376R/K376R* (n = 3). Scale bars, 50 μm. **e** Immunostaining for VE-cad (green) and cleaved caspase-3(CC3) (red) on yolk sacs from wild-type and *Ripk1K376R/K376R* embryos. Scale bars, 100 μm. Images are representative of embryos from E12.5 (n = 3/genotype). **f** The body samples from E12.5 embryos were analyzed by western blot and immunoblotted for PARP, CC3, p-RIPK1(S166), RIPK1, and RIPK3

embryos, indicating that the cell death induced by this mutation has effects on both embryonic tissues and yolk sacs (Fig. 1e). At E13.5 and E14.5, $Ripk1^{K376R/K376R}$ embryos were anemic with apparent developmental abnormalities which indicate the death of the mutant embryos (Fig. 1c). Therefore, these results suggest that germline mutation of $Ripk1^{K376R/K376R}$ causes embryonic lethality at E12.5 with excessive cell death including apoptosis and necroptosis.

**K376R blocks the ubiquitination of RIPK1**. To investigate the contribution of RIPK1 K376 on diverse signaling, we stimulated primary MEFs from $Ripk1^{K376R/K376R}$ mice and $Ripk1^{+/+}$ mice with TNF-α. Within 5 min of TNF-α stimulation, obvious ubiquitination of RIPK1 in complex I was detected in $Ripk1^{+/+}$ cells, whereas a significant reduction in RIPK1 ubiquitination was observed in $Ripk1^{K376R/K376R}$ cells (Fig. 2a). Accordingly, the interaction between RIPK1 and NEMO was markedly weakened in $Ripk1^{K376R/K376R}$ cells (Fig. 2c). Pre-treating cells with Smac mimetics, a cIAP antagonist, remarkably inhibited the ubiquitination of RIPK1 in response to TNF-α, suggesting that ubiquitination of RIPK1 on K376 was mainly mediated by E3 ubiquitin ligases cIAP1/2 (Fig. 2b). Immunoprecipitation of ubiquitin chains showed that K63 ubiquitination of RIPK1 was more profoundly affected by K376R mutation (Supplementary Fig. 1a), which was consistent with previous reports that showed the polyubiquitin chains on RIPK1 were linked primarily through K63 of ubiquitin[3,40]. To determine if the ubiquitination of RIPK1 on K376 was required for NF-κB activation, we performed a time-course experiment by treating cells with TNF-α. In $Ripk1^{+/+}$ cells, IκBα was phosphorylated within 5 min in response to TNF-α and was rapidly degraded (Fig. 2d). Consequently, phosphorylation and gradual translocation of p65 were observed (Fig. 2d, e), which activated the transcription of target genes (Fig. 2f). In contrast, in $Ripk1^{K376R/K376R}$ cells, IκBα activation was dramatically blocked, and subsequent translocation of p65 was delayed which led to impaired transcriptional induction of NF-κB-targeted genes (Fig. 2d–f). Importantly, except for NF-κB pathway, other signaling pathways in $Ripk1^{K376R/K376R}$ cells such as activation of JNK, p38, and Erk1/2 were almost unaffected (Fig. 2d), indicating a distinctive role of RIPK1$^{K376R}$ in activating NF-κB signaling in MEFs. Thus, our results show a critical role of K376 in mediating RIPK1 ubiquitination and NF-κB activation in response to TNF-α stimulation.

**RIPK1$^{K376R}$ promotes both apoptosis and necroptosis**. Defective polyubiquitination of RIPK1 has been proposed to promote TNF-α induced apoptosis in Jurkat T cells[3,5,38]. However, in MEFs, we found that the disruption of RIPK1 ubiquitination mediated by K376R sensitized cells to not only apoptosis but also necroptosis. In response to TNF-α, much more $Ripk1^{K376R/K376R}$ cells underwent cell death compared to $Ripk1^{+/+}$ control cells (Fig. 3a). After 24 h of TNF-α treatment, cleaved caspase-3 and phosphorylated MLKL were clearly detected in $Ripk1^{K376R/K376R}$ cells, indicating the simultaneous occurrence of apoptosis and necroptosis (Fig. 3b). In line with these results, mutant RIPK1$^{K376R}$ was found in complex II after TNF-α stimulation for 12 h, suggesting that RIPK1$^{K376R}$ promoted RIPK1 transition from complex I to complex II which led to the subsequent apoptosis and necroptosis (Fig. 3d). Besides, stimulating MEFs with TNF-α combined with Smac mimetics (TS), $Ripk1^{K376R/K376R}$ cells showed more sensitivity than the $Ripk1^{+/+}$ cells did (Fig. 3a). In the presence of caspase inhibitor zVAD-fmk (Z), TS induced higher level of necroptosis in $Ripk1^{K376R/K376R}$ MEFs compared with $Ripk1^{+/+}$ cells as the oligomerization of phosphorylated

MLKL and aggregation of RIPK1 and RIPK3 were detected in $Ripk1^{K376R/K376R}$ MEFs earlier than in $Ripk1^{+/+}$ MEFs (Fig. 3a, c).

We next investigated the effect of NF-κB signaling defects in $Ripk1^{K376R/K376R}$ MEFs on its sensitivity to cell death. Interestingly, activating NF-κB signaling by P65$^{S275D}$ overexpression failed to rescue mutant cells from TNF-α induced apoptosis and necroptosis (Supplementary Fig. 1b). Moreover, when treated with TNF-α combined with cycloheximide (CHX) which blocked NF-κB-dependent gene expression, $Ripk1^{K376R/K376R}$ MEFs were still more sensitive to cell death than the $Ripk1^{+/+}$ control (Fig. 3a). These results argued against NF-κB defects being the leading cause for enhanced cell death in mutant cells. Therefore, we conclude that RIPK1$^{K376R}$ promotes the formation of complex II to facilitate the induction of apoptosis and necroptosis.

**RIPK1$^{K376R}$ induced cell death relies on its kinase activity**. It has been shown that the activation of RIPK1 kinase activity, which promotes autophosphorylation of RIPK1, is critical for the transition from complex I to complex II[29]. Using phospho-Ser166-RIPK1 (p-RIPK1(S166)) antibody as a marker of RIPK1 activation, we found more phosphorylated RIPK1 in both complex I and complex II in TNF-α-stimulated $Ripk1^{K376R/K376R}$ MEFs (Figs. 2a, 3d). Thus, we wondered whether RIPK1$^{K376R}$-induced cell death relied on the kinase activity of RIPK1. Treatment of Necrostatin-1(Nec-1s or N), a RIPK1 kinase inhibitor, efficiently prolonged the survival of $Ripk1^{K376R/K376R}$ MEFs in response to TNF-α (or TC or TS) induced apoptosis and necroptosis (Fig. 4a). Similar results were obtained from treatments of TSZ or TCZ (Fig. 4a). Consistently, we found that the level of p-Ser166 RIPK1 was elevated in mutant cells and was significantly blocked by Nec-1s (Fig. 5c). Consequently, levels of cleaved caspase-3 and p-MLKL were reduced as well, indicating that RIPK1 ubiquitination on K376 is required for suppressing RIPK1 kinase-mediated the apoptosis and necroptosis (Fig. 5c).

As we observed higher levels of p-RIPK1 in $Ripk1^{K376R/K376R}$ embryos (Fig. 1e), we next investigated the effect of Nec-1s on the $Ripk1^{K376R/K376R}$ lethality in vivo. Feeding heterozygous mothers with Nec-1s during gestation period effectively prevented the lethal phenotypes of homozygous descendants (Fig. 4b). Compared with $Ripk1^{K376R/K376R}$ embryos without Nec-1s treatment which were dead and anemic at E13.5, Nec-1s-treated mutant embryos appeared much more normal although their yolk sacs were not completely rescued (Fig. 4b). Histological examination showed that Nec-1s treatment profoundly prevented the tissue losses in $Ripk1^{K376R/K376R}$ embryos (Fig. 4c). Besides, immunoblot analysis confirmed that RIPK1 phosphorylation and apoptosis featured by CC3 in mutant embryos were inhibited by Nec-1s treatment (Fig. 4f). Consistently, the number of TUNEL positive cells in the fetal livers of $Ripk1^{K376R/K376R}$ embryos was significantly reduced by Nec-1s treatment, suggesting that the massive cell death was largely restrained by RIPK1 kinase inhibition (Fig. 4d, e). Taken together, these results suggest that RIPK1$^{K376R}$-induced excessive cell death is primarily dependent on the kinase activity of RIPK1.

**$Ripk1^{K376R/-}$ mice are viable but develop severe inflammation**. Assuming that RIPK1 activation was responsible for the cell death in $Ripk1^{K376R/K376R}$ embryos, we next tested whether the amount of activated RIPK1 had an impact on this lethality. By crossing $Ripk1^{K376R/+}$ into $Ripk1^{+/-}$ mice, we generated $Ripk1^{K376R/-}$ mice with only half amounts of RIPK1$^{K376R}$. Strikingly, in contrast to $Ripk1^{K376R/K376R}$ mice which were totally abnormal at E13.5, $Ripk1^{K376R/-}$ mice developed normally (Figs. 1c, 5a). MEFs isolated from $Ripk1^{K376R/-}$ embryos were less sensitive to TNF-α induced apoptosis and necroptosis compared to $Ripk1^{K376R/K376R}$

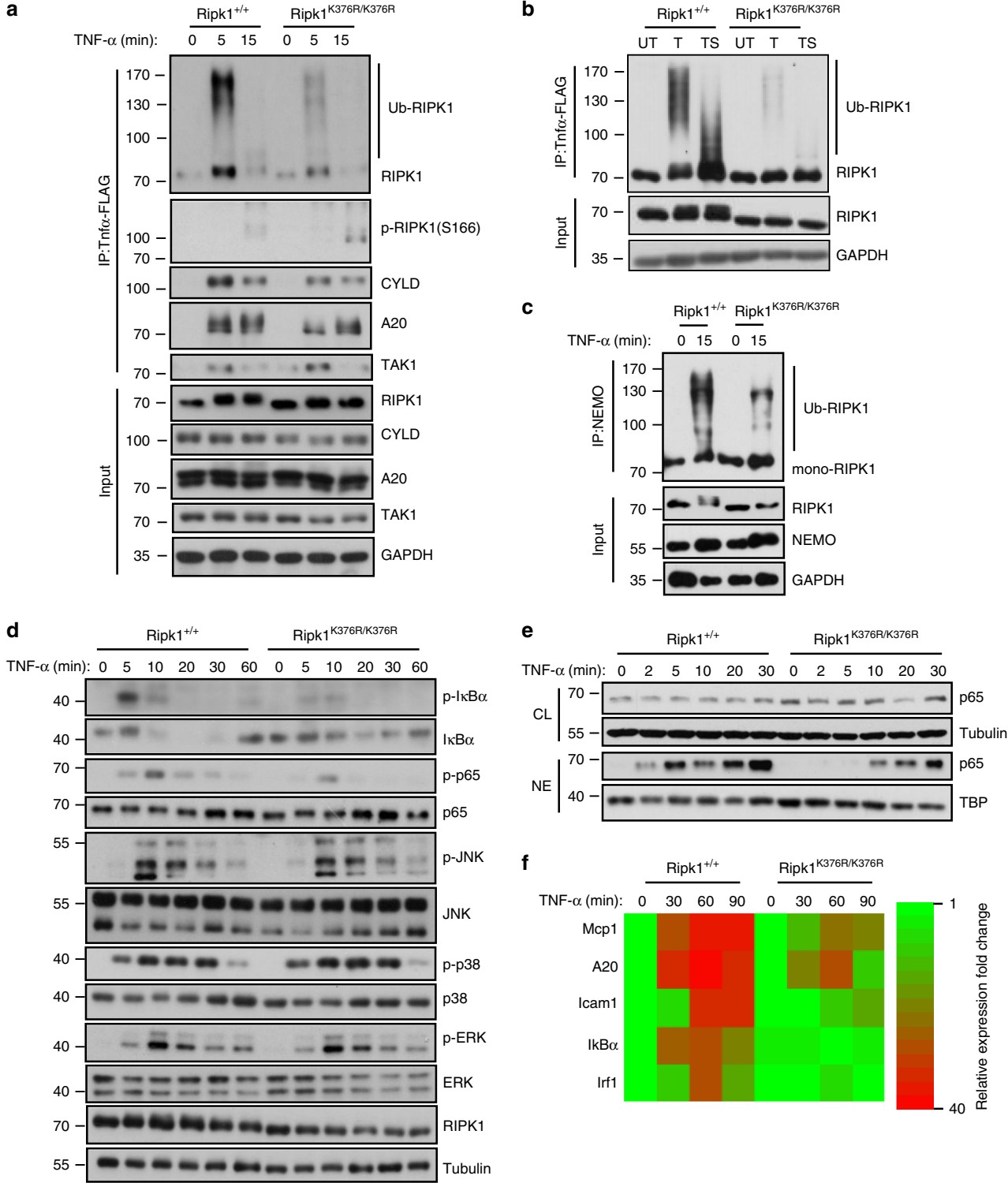

**Fig. 2** K376R blocks the ubiquitination of RIPK1 and NF-κB activation upon TNF-α stimulation. **a** *Ripk1*[+/+] and *Ripk1*[K376R/K376R] MEFs were treated with Flag-TNF-α (100 ng/ml) for the indicated periods of time, complex I was immnoprecipitated using anti-FLAG beads, endogenous RIPK1 ubiquitination and indicated proteins were detected by western blotting. **b** *Ripk1*[+/+] and *Ripk1*[K376R/K376R] MEFs were stimulated by Flag-TNF-α (100 ng/ml) for 5 min with or without pretreatment of Smac mimetics, complex I was immunoprecipitated using anti-FLAG beads, endogenous RIPK1 ubiquitination were detected by western blotting. Abbreviations are as follows: T Flag-TNF-α stimulation, TS Flag-TNF-α stimulation with Smac mimetics pretreatment. **c** *Ripk1*[+/+] and *Ripk1*[K376R/K376R] MEFs were treated with TNF-α (100 ng/ml) for 15 min and then NEMO was immunoprecipitated. Ubiquitinated RIPK1 was detected with RIP1 antibody. Cell lysates were analyzed by western blotting using the indicated antibodies. **d** *Ripk1*[+/+] and *Ripk1*[K376R/K376R] MEFs were treated with TNF-α (20 ng/ml) for the indicated periods of time, signaling pathways including NF-κB, JNKs, p38, and ERKs were examined by western blotting. **e** *Ripk1*[+/+] and *Ripk1*[K376R/K376R] MEFs were treated with TNF-α (20 ng/ml) for 5 min. P65 nuclear translocation was examined by western blotting. Abbreviations are as follows: NE nuclear extracts, CL cell lysates. **f** *Ripk1*[+/+] and *Ripk1*[K376R/K376R] MEFs were treated with TNF-α (20 ng/ml) for the indicated times, mRNA transcription change of NF-κB targeted genes were determined by quantitative PCR (qPCR)

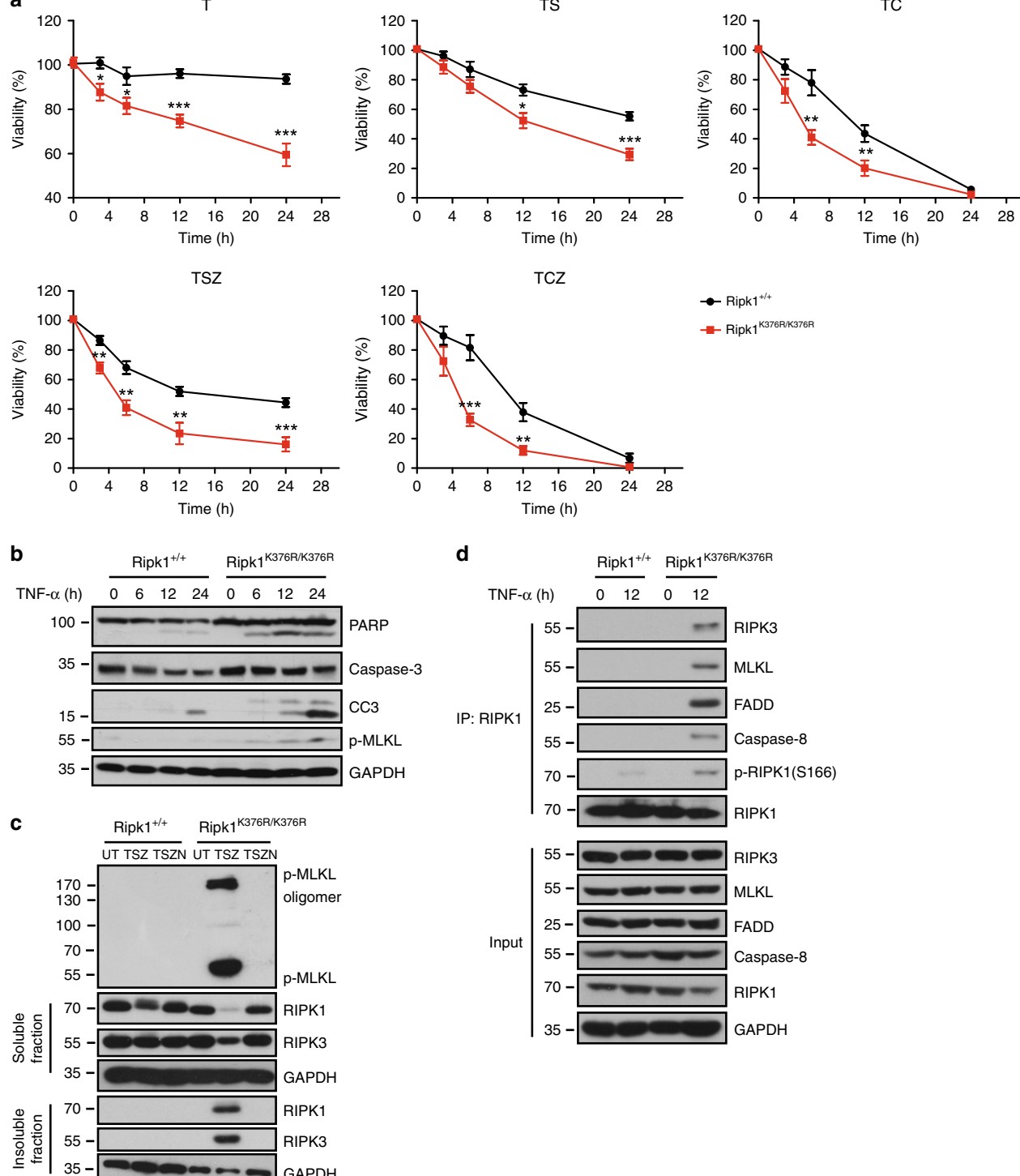

**Fig. 3** Cells bearing RIPK1$^{K376R/K376R}$ are more susceptible to TNF-α induced apoptosis and necroptosis. **a** *Ripk1*$^{+/+}$ and *Ripk1*$^{K376R/K376R}$ MEFs were treated with T, TS, TSZ, TC, TCZ for indicated times, respectively. Cell viability was determined using the CellTiter-Glo kit. The data are represented as the mean ± SEM of MEFs derived from 3 embryos. *P* values were determined by Student's *t*-test (*$p < 0.05$, **$p < 0.01$, ***$p < 0.001$). Abbreviations are as follows: UT untreated, TSZ TNF-α (20 ng/ml)+Smac mimetic(100 nM) +zVAD(20 μM); TSZN, TNF-α+Smac mimetic +zVAD+Necrostatin-1 (20 μM); TC, TNF-α+CHX (20 ng/ml); TCZ, TNF-α+CHX +zVAD. Source data are provided as a Source Data file. **b** *Ripk1*$^{+/+}$ and *Ripk1*$^{K376R/K376R}$ MEFs were treated with mouse TNF-α (20 ng/ml) for indicated times, the cell lysates were analyzed by western blotting using the indicated antibodies. **c** *Ripk1*$^{+/+}$ and *Ripk1*$^{K376R/K376R}$ MEFs were treated with TSZ or TSZN for 3 h, the cell lysates were resolved on non-reducing gel and immunoblotted with anti-p-MLKL antibody. **d** *Ripk1*$^{+/+}$ and *Ripk1*$^{K376R/K376R}$ MEFs were treated with mouse TNF-α (20 ng/ml) for 12 h, anti-RIPK1 was used to immunoprecipitate complex and immunocomplexes were analyzed by western blotting using indicated antibody

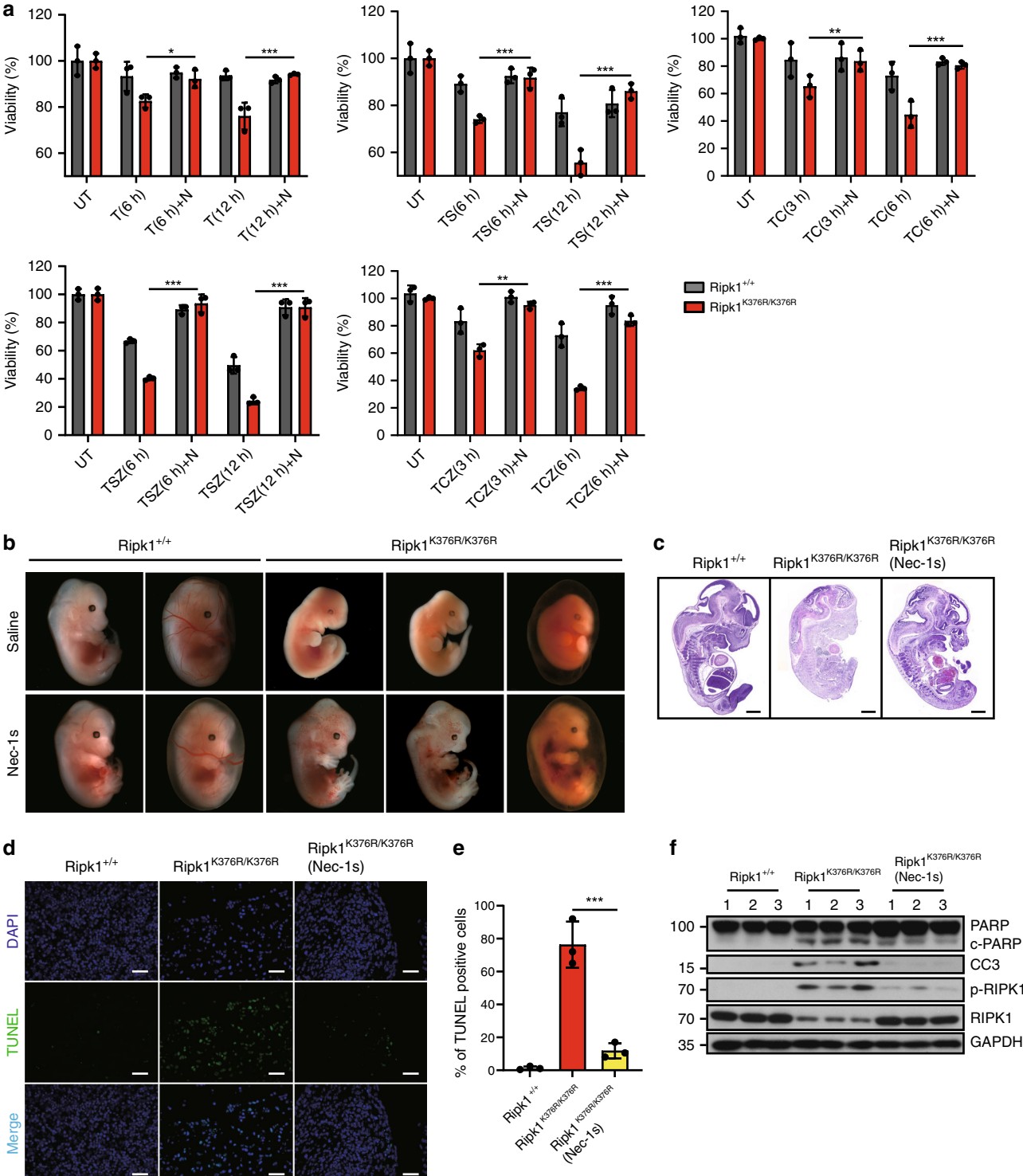

**Fig. 4** Treatment of Nec-1s inhibits cell death induced by RIPK1 K376R. **a** *Ripk1*[+/+] and *Ripk1*[K376R/K376R] MEFs were treated with T, TS, TSZ, TC, TCZ for indicated times with or without Nec-1s as indicated. Cell viability was determined using the CellTiter-Glo kit. The data are represented as the mean ± SEM (n = 3). *P* values were determined by Student's *t*-test (*p < 0.05, **p < 0.01, ***p < 0.001). Source data are provided as a Source Data file. **b** Pregnant *Ripk1*[K376R/+] females were fed with Nec-1s (50 mg/kg) or saline solution once per day since day 9.5 of gestation and sacrificed on day 13.5. Representative embryos with indicated genotypes were presented. **c** Representative H&E staining (scale bars, 1 mm) and **d** TUNEL staining of fetal livers from mouse embryos of the indicated genotypes. Each image is representative of at least three embryos. Scale bars, 50 μm. **e** Quantification of TUNEL positive cells in Fig. 4d. The results are represented as ± SEM of three embryos per group. *P* value was calculated by Student's *t*-test (***p < 0.001). Source data are provided as a Source Data file. **f** The fetal liver samples from three embryos per group were analyzed by western blot and immunoblotted with PARP, CC3, p-RIPK1(S166), and RIPK1

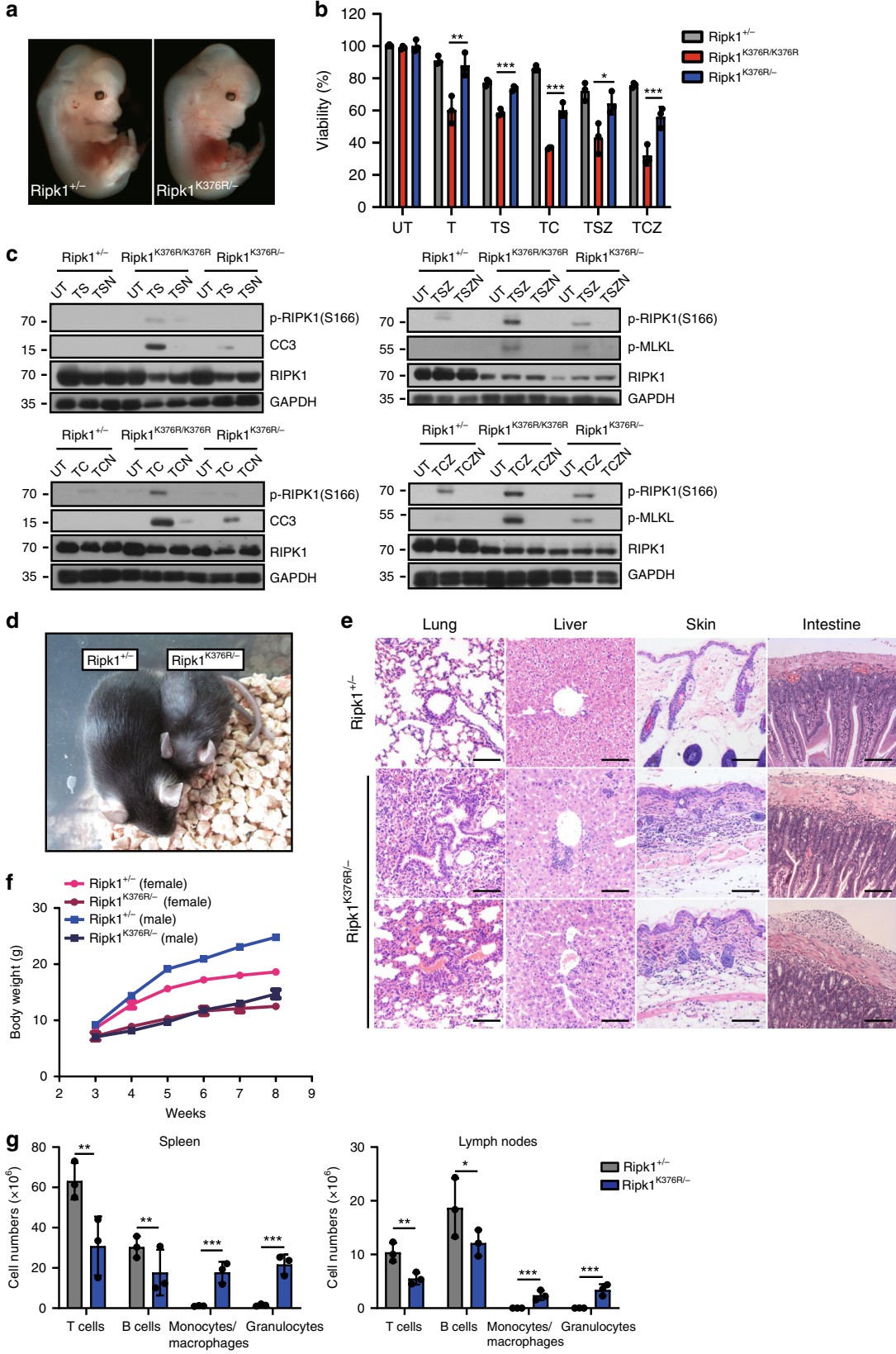

MEFs (Fig. 5b). Similar results were obtained when treating cells with TC, TS, TCZ, and TSZ (Fig. 5b). In line with these data, the high levels of p-RIPK1(S166) with cleaved caspase-3 and phosphorylation of MLKL detected in *Ripk1*^K376R/K376R MEFs were profoundly reduced in *Ripk1*^K376R/− MEFs (Fig. 5c). These results

suggest that there is likely to be a threshold above which activated RIPK1^K376R can trigger the lethal pathways during embryogenesis.

Intriguingly, *Ripk1*^K376R/− mice can grow up into adulthood, although they displayed much smaller and had lower body weight than their littermate controls (Fig. 5d, f). Histological

**Fig. 5** $Ripk1^{K376R/-}$ mice are viable but display systemic inflammation. **a** Representative whole-mount dark field images of E13.5 embryos with indicated genotypes. **b** $Ripk1^{+/-}$, $Ripk1^{K376R/K376R}$, and $Ripk1^{K376R/-}$ MEFs were treated with T, TS, TSZ, TC, and TCZ, respectively. Cell viability was determined using the CellTiter-Glo kit. Data are represented as mean ± SEM of MEFs derived from three embryos. P values were determined by Student's t-test (*p < 0.05, **p < 0.01, ***p < 0.001). Abbreviations are as follows: UT untreated, TSZ TNF-α (20 ng/ml)+Smac mimetic(100nM) +zVAD(20 μM), TSZN TNF-α +Smac mimetic +zVAD+Necrostatin-1(20 μM), TC TNF-α+CHX(20 ng/ml), TCZ TNF-α+CHX +zVAD. Source data are provided as a Source Data file. **c** $Ripk1^{+/-}$, $Ripk1^{K376R/K376R}$, and $Ripk1^{K376R/-}$ MEFs were treated with TS(6h), TSZ(2h), TC(3h), TCZ(1h) with or without Nec-1s. Western blottings of indicated proteins were shown. **d** Representative 8-week-old $Ripk1^{K376R/K376R}$ mouse and littermate control. **e** Representative hematoxylin and eosin (H&E)-stained tissues from 4-week-old animals of the indicated genotypes. (Scale bars, 100 μm). **f** Plot of weight of littermate $Ripk1^{+/-}$ and $Ripk1^{K376R/-}$ mice. **g** Different cell subsets in spleens and LNs from 6-week-old mice with indicated genotypes were determined by flow cytometry using following markers: B cells (CD19$^+$ or B220$^+$), T cells (CD3$^+$), Monocytes and macrophages (CD11b$^+$), Granulocytes (Gr-1$^+$). Data are represented as mean ± SEM (n = 3/genotype). P values were calculated by Student's t-test (*p < 0.05, **p < 0.01, ***p < 0.001). Source data are provided as a Source Data file

examination of 4-week-old to 8-week-old $Ripk1^{K376R/-}$ mice revealed signs of inflammation in multiple organs including skins, lungs, intestines, and livers (Fig. 5e). Flow cytometric analysis of $Ripk1^{K376R/-}$ spleens and lymph nodes revealed decreased numbers of T cells (CD3$^+$) and B cells (B220$^+$ or CD19$^+$) while remarkably increased numbers of granulocytes (Gr-1$^+$) and macrophages (CD11b$^+$) (Fig. 5g), indicating that $Ripk1^{K376R/-}$ mice developed systemic inflammation after birth. Collectively, these data suggest that K376-mediated RIPK1 ubiquitination is critical for restraining inflammation during postnatal development.

**Loss of *Tnfr1* rescues the lethality of $Ripk1^{K376R/K376R}$ mice**. To further investigate the upstream signaling inducing the $Ripk1^{K376R/K376R}$ lethality, we crossed $Ripk1^{K376R}$ into $Tnfr1^{-/-}$ or $Ifnar^{-/-}$ background to generate $Tnfr1^{-/-}$ $Ripk1^{K376R/K376R}$ and $Ifnar^{-/-}Ripk1^{K376R/K376R}$ mice, respectively. Results from timed pregnancies showed that deletion of $Tnfr1$ fully rescued the $Ripk1^{K376R/K376R}$ lethality during embryonic development (Fig. 6a–c). In contrast, ablation of $Ifnar$ had no effect on the $Ripk1^{K376R/K376R}$ embryos and failed to rescue them from embryonic lethality (Fig. 6a). Consistent with these observations, the numbers of TUNEL-positive cells were remarkably increased and cleaved caspase-3 was clearly detected in the fetal livers of $Ifnar^{-/-}Ripk1^{K376R/K376R}$ embryos (Supplementary Fig. 2a, b). Thus, these results suggest that TNFR1, rather than IFNR1, induces signaling that triggers the death of $Ripk1^{K376R/K376R}$ mice during embryogenesis.

Although ablation of $Tnfr1$ prevented the embryonic death of $Ripk1^{K376R/K376R}$ mice, $Tnfr1^{-/-}Ripk1^{K376R/K376R}$ mice died around 2 weeks after birth (Fig. 6h). These mice succumb to systemic inflammation as obvious inflammatory cell infiltrations were observed in the skin, lung, liver, and intestine (Fig. 6e). Quantitative RT-PCR examination revealed enhanced levels of inflammatory cytokines in the lungs and intestines of $Tnfr1^{-/-}$ $Ripk1^{K376R/K376R}$ mice (Fig. 6d). In addition, the composition of immune cells in spleen and thymus of mutant mice was largely disrupted. Compared with $Tnfr1^{-/-}$ controls, the numbers of lymphocytes (CD3$^+$ cells and B220$^+$ cells) in the spleen were reduced, whereas the numbers of myeloid cells (CD11b$^+$ or Gr-1$^+$ cells) were greatly increased (Supplementary Fig. 2d, e). In the thymus, the total cell counts were dramatically reduced with abnormal compartments of T cells in which CD4$^+$ and CD8$^+$ single-positive populations were augmented while CD4$^+$CD8$^+$ double-positive population was decreased (Supplementary Fig. 2d, e). These results showed that the homeostasis of the immune system in $Tnfr1^{-/-}Ripk1^{K376R/K376R}$ mice was disrupted, suggesting that K376-mediated ubiquitination of RIPK1 also contributes to regulating immune homeostasis.

We then sought to address the cause for the postnatal death of $Tnfr1^{-/-}Ripk1^{K376R/K376R}$. Histological examination showed multiple tissues from $Tnfr1^{-/-}Ripk1^{K376R/K376R}$ mice displayed

tissue loss and abnormal architecture, indicating that these phenotypes may be a result of massive cell death (Fig. 6e). Using cleaved caspase-3 as a marker for apoptosis, we observed no significantly positive cells detected in skins, livers and intestines of $Tnfr1^{-/-}Ripk1^{K376R/K376R}$ mice, suggesting that apoptosis was not the major cause for the phenotypes (Supplementary Fig. 2c). However, when we treated bone marrow-derived macrophages (BMDMs) from the $Tnfr1^{-/-}Ripk1^{K376R/K376R}$ mice with LPS or Poly(I:C) in the presence of zVAD, the cells were much more vulnerable compared to their controls and the phosphorylations of RIPK1 and RIPK3 were more significantly induced in these cells, suggesting that RIPK3-mediated necroptosis plays an important role in this circumstance (Supplementary Fig. 2f, g). Therefore, we tested the contribution of RIPK3-mediated signaling by crossing $Ripk3^{-/-}$ into $Tnfr1^{-/-}Ripk1^{K376R/K376R}$ mice. Importantly, deletion of Ripk3 rescued the postnatal death of $Tnfr1^{-/-}Ripk1^{K376R/K376R}$ and allowed them normal development (Fig. 6f, h). Moreover, histological examination indicated that Ripk3 ablation markedly prevented the tissue loss and inflammation in $Tnfr1^{-/-}Ripk1^{K376R/K376R}$ mice (Fig. 6g). Taken together, these findings reveal a critical role of K376-meidated RIPK1 ubiquitination in restraining RIPK3-dependent inflammation during postnatal development.

**Ablation of *Fadd* and *Ripk3*/*Mlkl* rescues $Ripk1^{K376R/K376R}$ mice**. Given that the activation of RIPK1 led to two subsequent signaling pathways including FADD/Caspase-8-mediated apoptosis and RIPK3/MLKL-mediated necroptosis, we next asked whether $Ripk1^{K376R/K376R}$ lethality was caused by these pathways. First, we crossed $Ripk1^{K376R/+}$ into $Ripk3^{-/-}$ or $Mlkl^{-/-}$ mice to test the contribution of necroptosis. The resulting $Ripk3^{-/-}Ripk1^{K376R/K376R}$ and $Mlkl^{-/-}Ripk1^{K376R/K376R}$ mice died during embryogenesis as we observed no homozygous descendant from heterozygous breeding (Supplementary Fig. 3b). The results from timed pregnancies showed that $Ripk3^{-/-}Ripk1^{K376R/K376R}$ and $Mlkl^{-/-}$ $Ripk1^{K376R/K376R}$ embryos appeared normal at E13.5 but displayed apparent developmental abnormalities at E14.5 (Supplementary Fig. 3a). Although ablation of Ripk3 or Mlkl slightly prolonged the survival of $Ripk1^{K376R/K376R}$ embryos, the lethality was not prevented. These findings indicate the involvement of RIPK3/MLKL-independent pathways.

Next, we bred $Ripk1^{K376R/+}$ into $Ripk3^{-/-}Fadd^{-/-}$ and $Mlkl^{-/-}Fadd^{-/-}$ mice to generate $Ripk3^{-/-}Fadd^{-/-}Ripk1^{K376R/K376R}$ and $Mlkl^{-/-}Fadd^{-/-}Ripk1^{K376R/K376R}$ mice, respectively. Ablation of Fadd and Ripk3 or Mlkl fully rescued the embryonic lethality of $Ripk1^{K376R/K376R}$ mice and displayed normal development and maturation (Fig. 7a, b). MEFs isolated from these triple mutant mice were resistant to TNF-α triggered apoptosis and necroptosis while were defective in activating NF-κB signaling as $Ripk1^{K376R/K376R}$ cells, suggesting that massive cell death rather than impaired NF-κB activation contributes to the lethality of $Ripk1^{K376R/K376R}$ mice (Supplementary Fig. 4b, c). RIPK1$^{K376R}$ was widely detected in the

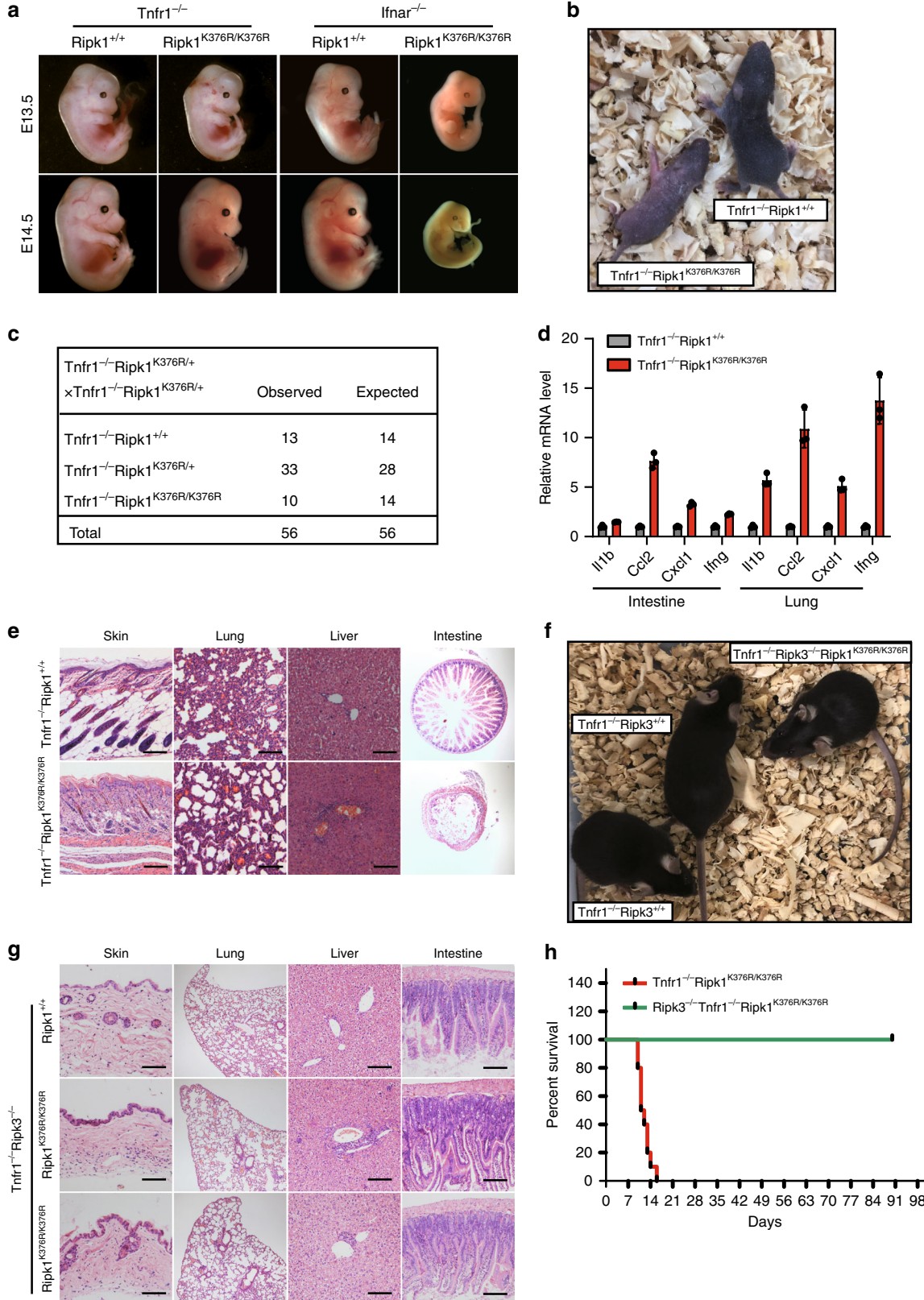

major organs of triple-mutant mice and its expression levels were comparable to those of WT controls, suggesting that RIPK1[K376R] did not affect the expression of RIPK1 (Supplementary Fig. 4a). When aged, *Ripk3*[−/−]*Fadd*[−/−]*Ripk1*[K376R/K376R] and *Mlkl*[−/−]*Fadd*[−/−]*Ripk1*[K376R/K376R] mice developed lymphoproliferative syndrome with an accumulation of CD3[+]B220[+] lymphocytes

in peripheral lymphoid organs seen in *Ripk3*[−/−]*Fadd*[−/−] and *Mlkl*[−/−]*Fadd*[−/−] mice (Fig. 7d, f, Supplementary Fig. 4d)[19–22,41–43]. The disease was characterized by lymphadenopathy and splenomegaly, which disrupted the normal architecture of the lymph nodes and spleen (Fig. 7c, d). Flow cytometric analysis revealed profoundly increased numbers of multiple cells in spleen and lymph nodes,

**Fig. 6** $Tnfr1^{-/-}Ripk1^{K376R/K376R}$ mice die postnatally due to RIPK3-dependent inflammation. **a** Whole-mount dark field images of embryos with indicated genotypes. Images are representative of embryos from E13.5 ($n = 8$) and E14.5 ($n = 9$). **b** Representative 1-week-old $Tnfr1^{-/-}Ripk1^{K376R/K376R}$ mice and littermate control. **c** Expected and observed frequency of genotypes in offspring at weaning from crosses of $Tnfr1^{-/-}Ripk1^{K376R/+}$. **d** Quantitative RT-PCR analysis of the mRNA expression of cytokines in lungs and intestines from 1-week-old pups with indicated genotypes. Data are represented as mean ± SEM ($n = 3$). Source data are provided as a Source Data file. **e** Representative H&E stained tissues from 1-week-old $Tnfr1^{-/-}Ripk1^{K376R/K376R}$ and littermate control. Scale bars, 100 μm. **f** Representative 8-week-old $Tnfr1^{-/-}Ripk3^{-/-}Ripk1^{K376R/K376R}$ and its littermate controls. **g** Representative H&E stained tissues from 8-week-old $Tnfr1^{-/-}Ripk3^{-/-}Ripk1^{K376R/K376R}$ and littermate control. Scale bars, 100 μm. **h** Kaplan-Meier plot of mouse survival from intercrosses of $Tnfr1^{-/-}Ripk1^{K376R/+}Ripk3^{+/-}$

including lymphocytes (T cells/B cells), macrophages and granulocytes (Fig. 7e). In addition, the inflammation was observed in multiple tissues of $Ripk3^{-/-}Fadd^{-/-}Ripk1^{K376R/K376R}$ or $Mlkl^{-/-}Fadd^{-/-}Ripk1^{K376R/K376R}$ mice, indicating the occurrence of systemic inflammation (Fig. 7c). Collectively, these data indicate that blockade of both FADD-mediated apoptotic pathway and RIPK3/MLKL-mediated necroptotic pathway rescues the embryonic lethality of $Ripk1^{K376R/K376R}$ mice, although these mice develop lymphoproliferative diseases with systemic inflammation.

## Discussion

In this report, we show that the disruption of RIPK1 ubiquitination in vivo by a point mutation at K376 causes embryonic lethality at E12.5, which is completely different from the observation in $Ripk1$ deficient mice. A fundamental question emerged from this result: why does germline $Ripk1^{K376R}$ mutation cause lethality during embryogenesis while deletion of $Ripk1$ does not? The answer to the question may come from the unique structure of RIPK1 which contains a C-terminal death domain (DD), an intermediate domain harboring a RHIM and an N-terminal Ser/Thr kinase domain. The DD and RHIM can, respectively, recruit FADD and RIPK3 to initiate extrinsic apoptosis and necroptosis, while the intermediate domain can mediate the activation of NF-κB signaling[28,44,45]. In our study, we substituted K376 of the intermediate domain with R376 to block polyubiquitination of RIPK1. This mutation leads to two consequences: the defective NF-κB activation and the excessive cell death. These findings allowed two conclusions: (1) K376-mediated RIPK1 ubiquitination is responsible to activate NF-κB in vivo and (2) K376-mediated RIPK1 ubiquitination is critical to suppress apoptosis and necroptosis during development. In fact, RIPK1$^{K376R}$ mutation does not affect the expression of RIPK1 and functions of other domains, but was prone to the activation of RIPK1. This can be prevented by RIPK1 kinase inhibitor Nec-1 or deletion of one $Ripk1^{K376R}$ allele, indicating that the excessive RIPK1 activation which subsequently causes the apoptosis and necroptosis leads to the lethality during embryogenesis. Therefore, RIPK1 ubiquitination restrains RIPK1 itself from activating cell death during embryonic development, and that's why deletion of $Ripk1$ does not lead to lethal phenotypes at this stage.

$Ripk1^{-/-}$ mice die soon after birth, this lethality is initially thought to be due to the loss of the pro-survival function of RIPK1 which mediates NF-κB signaling[46]. The contribution of NF-κB signaling to maintain normal embryonic development has been investigated previously, and mice lacking the components of NF-κB die during embryogenesis due to massive cell death in multiple tissues[47–49]. Moreover, inactivating RIPK1 kinase can rescue the embryonic lethality of $RelA$-deficient mice[50,51]. The phenotypes of these mice resemble what we observed in $Ripk1^{K376R/K376R}$ mice, indicating that impaired NF-κB signaling may be a cause that leads to the embryonic lethality of $Ripk1^{K376R/K376R}$ mice. However, when treating $Ripk1^{K376R/K376R}$ cells with TNF-α combined with CHX which blocked NF-κB-dependent gene expression, $Ripk1^{K376R/K376R}$ MEFs were still

more vulnerable to the stimuli. Furthermore, activating NF-κB signaling by p65$^{S275D}$ overexpression failed to prevent the cell death of mutant cells. Besides, it has been reported that RIPK1 K377-mediated (human RIPK1) inhibition of TNF-α induced apoptosis is independent on NF-κB at the early stage of stimulation[38], which is consistent with what we observed in $Ripk1^{K376R/K376R}$ cells. Therefore, NF-κB defects might not be the leading cause of $Ripk1^{K376R/K376R}$ lethality.

The perinatal death of $Ripk1^{-/-}$ mice was fully rescued by ablation of both $Caspase-8/Fadd$ and $Ripk3$[19–22]. Thus, RIPK1 was considered to suppress lethal pathways engaged by FADD-Caspase-8 and RIPK3. However, the mechanism by which RIPK1 suppresses the perinatal lethality remains elusive. The mutation of RHIM in RIPK1 causes perinatal death similar to the observations from $Ripk1^{-/-}$ mice, and this lethality can be rescued by ablation of $Ripk3$ or $Zbp1$[35,36]. These investigations suggest that RIPK1 suppresses the perinatal necroptosis mediated by RIPK3/ZBP1 through RHIM domain. In our study, ablation of $Ripk3$ rescues the postnatal death of $Tnfr1^{-/-}Ripk1^{K376R/K376R}$ mice. Thus, we assume that RIPK1 K376 is important for restraining RIPK3-dependent necroptosis and ensuing inflammation after birth. Intriguingly, in the settings of RIPK1$^{K376R}$ mutation, RIPK1 fails to prevent the postnatal necroptosis through RHIM, suggesting that K376-mediated ubiquitination may regulate RHIM-mediated suppression of RIPK3-dependent necroptosis.

Although ablation of $Tnfr1$ rescued $Ripk1^{K376R/K376R}$ mice during embryonic development, $Tnfr1^{-/-}Ripk1^{K376R/K376R}$ mice died postnatally. These results indicate RIPK1 ubiquitination on K376 may mediate divergent pathways at different stages of development. TNF/TNFR1 is the main upstream signal of K376-mediated pathways during embryonic development, while additional signals are involved after birth. Although it is not clear which signaling pathways participated, the investigations on $Ripk1^{-/-}$ mice could provide some candidates. Similar to our findings in $Tnfr1^{-/-}Ripk1^{K376R/K376R}$ mice, $Tnfr1^{-/-}Ripk1^{-/-}$ mice died within 2 weeks after birth[21]. Interestingly, deletion of $Trif$ or type I interferon receptor prolonged the lifespan of $Ripk1^{-/-}Tnfr1^{-/-}$ mice, although none of the mice survived past weaning[21]. In addition, deletion of MyD88 improved the survival time of $Ripk1^{-/-}$ mice[22]. Therefore, we assume that, in addition to TNFR1, IFNR1, and TRIF are of great possibility to trigger the RIPK1 K376-meidated pathways. Intriguingly, as ablation of Ripk3 rescues the lethality of $Tnfr1^{-/-}Ripk1K376R/K376R$ mice, the downstream signaling mediating the lethal phenotypes is dependent on RIPK3. Recent studies have implied that RIPK3 has more functions beyond mediating necroptosis, therefore, additional investigations on examining the contribution of MLKL in this signaling pathway are still required.

By changing its ubiquitination status in response to TNF-α, RIPK1 can transduce either a pro-survival or death signal. Germline mutation of RIPK1 K376 leads to embryonic lethality at E12.5, which was prevented by deletion of $Tnfr1$, though $Tnfr1^{-/-}$ $Ripk1^{K376R/K376R}$ mice died of RIPK3-mediated inflammation after birth. Remarkably, one copy of RIPK1 is

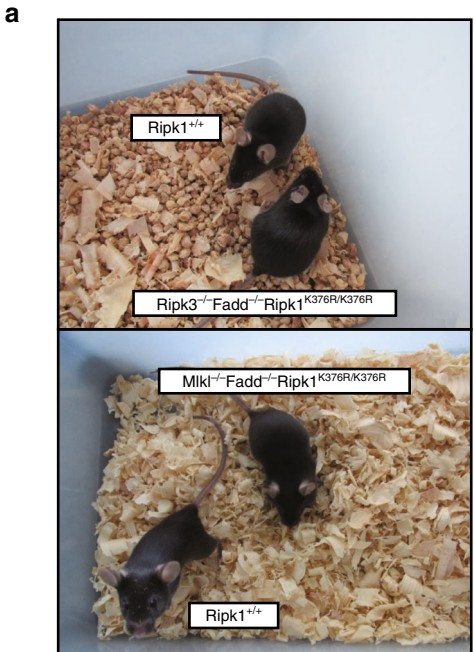

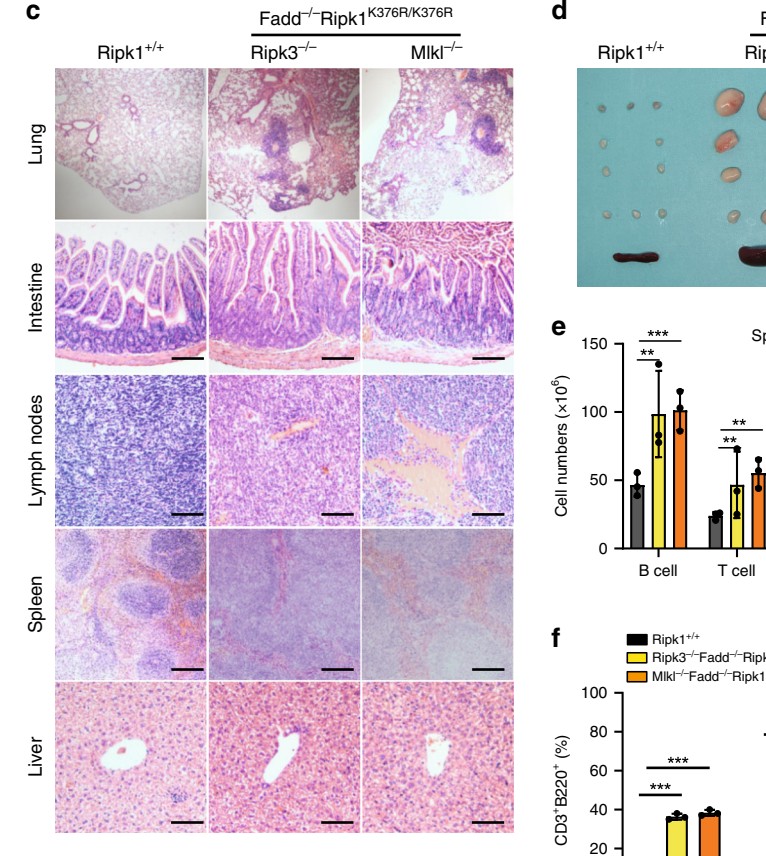

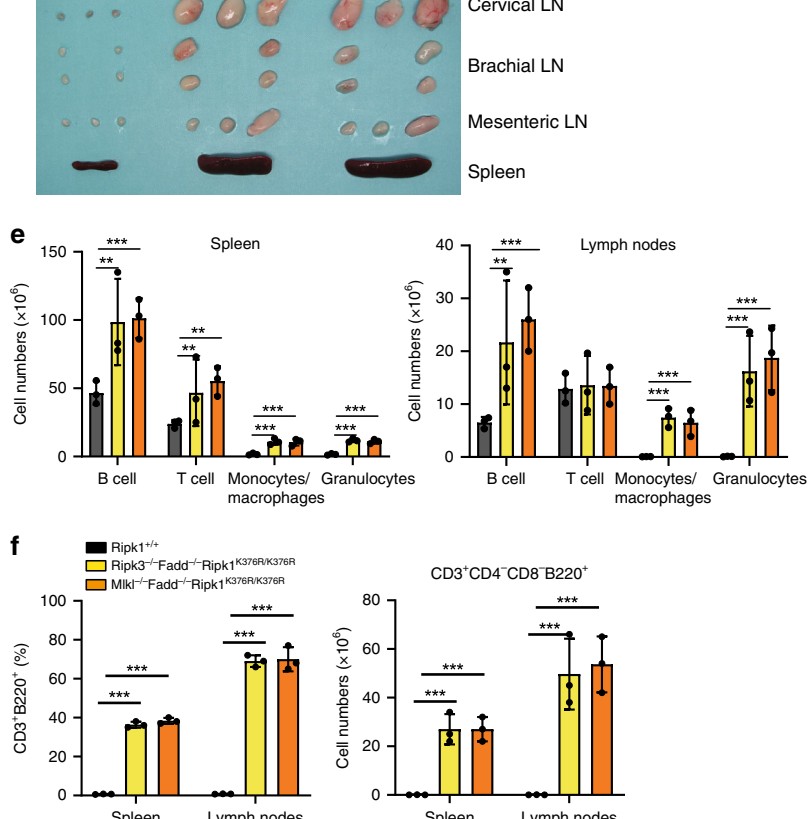

**b**

| Ripk3⁻/⁻Fadd⁻/⁻Ripk1^(K376R/+) × Ripk3⁻/⁻Fadd⁻/⁻Ripk1^(K376R/+) | Expected | Observed |
|---|---|---|
| Ripk3⁻/⁻Fadd⁺/⁻Ripk1^(+/+) | 9 | 11 |
| Ripk3⁻/⁻Fadd⁺/⁻Ripk1^(K376R/+) | 18 | 20 |
| Ripk3⁻/⁻Fadd⁺/⁻Ripk1^(K376R/K376R) | 9 | 0 |
| Ripk3⁻/⁻Fadd⁻/⁻Ripk1^(+/+) | 9 | 12 |
| Ripk3⁻/⁻Fadd⁻/⁻Ripk1^(K376R/+) | 18 | 21 |
| Ripk3⁻/⁻Fadd⁻/⁻Ripk1^(K376R/K376R) | 9 | 8 |
| Total | 72 | 72 |
| Mlkl⁻/⁻Fadd⁻/⁻Ripk1^(K376R/+) × Mlkl⁻/⁻Fadd⁺/⁻Ripk1^(K376R/+) | Expected | Observed |
| Mlkl⁻/⁻Fadd⁺/⁻Ripk1^(+/+) | 8 | 10 |
| Mlkl⁻/⁻Fadd⁺/⁻Ripk1^(K376R/+) | 16 | 18 |
| Mlkl⁻/⁻Fadd⁺/⁻Ripk1^(K376R/K376R) | 8 | 0 |
| Mlkl⁻/⁻Fadd⁻/⁻Ripk1^(+/+) | 8 | 10 |
| Mlkl⁻/⁻Fadd⁻/⁻Ripk1^(K376R/+) | 16 | 19 |
| Mlkl⁻/⁻Fadd⁻/⁻Ripk1^(K376R/K376R) | 8 | 7 |
| Total | 64 | 64 |

required to prevent a *Fadd/Ripk3* or *Fadd/Mlkl* dependent lethality in *Ripk1^(K376R/K376R)* mice, possibly relating to a RIPK1 activation threshold (Supplementary Fig. 5). These genetic evidences provide a comprehensive understanding of the in vivo function of RIPK1 ubiquitination, which is essential for both embryonic development and immune homeostasis.

Importantly, these findings can help to explain some seemingly contradictory results from previously reported genetic studies. Furthermore, a more recent study has highlighted the role of RIPK1 in human disease assocoiated with immunodeficiency, arthritis, and intestinal inflammation, which confirmed the function of RIPK1 in humans immune system[52]. Therefore, the

**Fig. 7** Embryonic lethality of *Ripk1*$^{K376R/K376R}$ mice is fully rescued by ablation of *Fadd* and *Ripk3/Mlkl*. **a** Representative photographs of mice with the indicated genotypes. **b** Predicted and observed frequencies in offspring from crosses of animals with the indicated genotypes. **c** Representative H&E-stained tissues from 12-week-old animals of the indicated genotypes. Results are representative of three mice of each genotype (Scale bars, 100 μm). **d** Representative lymphoid organs removed from 12-week-old mice of indicated genotypes. **e** Different cell subsets from 12-week-old animals were determined by flow cytometry using following markers: B cells (CD19+ or B220+), T cells(CD3+), Monocytes and macrophages(CD11b+), Granulocytes (Gr-1+). Data are represented as mean ± SEM (n = 3/genotype). P values were determined by Student's t-test (**p < 0.01, ***p < 0.001). **f** Percentages and cell numbers of CD3+B220+ lymphocytes in spleen and lymph nodes from 12-week-old mice of indicated genotypes. Data are represented as mean ± SEM (n = 3/genotype). P values were determined by Student's t-test (**p < 0.01, ***p < 0.001). Source data are provided as a Source Data file

potential role of RIPK1 ubiquitination in regulating specific signaling pathways other than the TNF-α mediated pathway will be of great interest for providing new therapeutic opportunities in related human diseases, and mice bearing RIPK1$^{K376R}$ mutation can be used to explore the contribution of RIPK1 ubiquitination to a variety of mouse models of human diseases.

## Methods

**Mice**. Mice were housed in a specific pathogen-free (SPF) facility. *Ripk3*$^{−/−}$, *Fadd*$^{+/−}$ and *Mlkl*$^{−/−}$, and *Ifnar*$^{−/−}$ mouse lines have been described earlier[41,53]. *Ripk1*$^{K376R/+}$ and *Ripk1*$^{+/−}$ mice were generated by CRISPR-Cas9 mutation system (Bioray Laboratories Inc., Shanghai, China). The mouse *Ripk1*$^{K376R}$ mutation construct corresponds to the following genomic position (NC_000079.6) chr13: 34027834–34027836. *Ripk1*$^{K376R/+}$ mouse genotyping primers: 5′ CCATCTCCA GCCCTCCTA and 5′ TGCACTGCAATTCCACGA amplified 690 bp DNA fragments for sequencing. *Ripk1*$^{+/−}$ mice were generated by a 101 bp deletion in the coding region of exon 2 using CRISPR-Cas9 mutation system with sgRNA (5′-GACCTAGACAGCGGAGGCTT-3′). Ripk1$^{+/−}$ mice genotyping primers (RIPK1-F: 5′-GTGGTACTTTGGGAGGTGGAGG-3′ and RIPK1-R: 5′-CAGGATGACA AATCCATGGCTTCTG-3′) amplified 430 bp wild-type, 329 bp knock-out DNA fragments. Additional information is provided upon request. All mice utilized in this study have been backcrossed onto the C57BL/6 background for more than eight generations. Animal experiments were conducted in accordance with the guidelines of the Institutional Animal Care and Use Committee of the Institute of Nutrition and Health, Shanghai Institutes for Biological Sciences, University of Chinese Academy of Sciences.

**Reagents**. The Smac mimetic, Cycloheximide and LPS were purchased from Sigma. TNF-α and z-VAD were from R&D (Minneapolis, MN, USA) and Calbiochem (Anaheim, CA, USA), respectively. Necrostatin-1 was purchased from Enzo Life Science (Alexis, USA). The following antibodies were used for western blotting and immunoprecipitation experiments: RIPK1 (1:2000, BD Biosciences, 610459), RIPK1 (1:2000, Cell Signaling Technology, 3493P), p-RIPK1(1:1000, a gift from Junying Yuan's lab), RIPK3 (1:5000, Prosci, 2283), p-RIPK3(1:1000. Abcam, ab195117), Caspase-8 (1:2000, Enzo Life Science, ALX-804-447-C100), MLKL (1:2000, Abgent, ap14272b), p-MLKL (1:1000, Abcam, ab196436), ß-actin (1:5000, Sigma, A3854), Tublin (1:5000, Sigma, T6199), IkBα (1:2000, Cell Signaling Technology, 9242S), p-IkBα (1:1000, Cell Signaling Technology, 9246S), p-ERK (1:2000, Cell Signaling Technology, 9101S), ERK (1:2000. Cell Signaling Technology,9102S), p-p38 (1:2000, Cell Signaling Technology, 9211S), p38 (1:2000, Cell Signaling Technology, 9228S), p-p65 (1:2000, Cell Signaling Technology, 3033S), p65 (1:2000, Cell Signaling Technology, 4764S), p-JNK (1:1000, Cell Signaling Technology, 9251S), JNK (1:2000, Cell Signaling Technology, 9252S), PARP (1:2000,Cell Signaling Technology, 9542S), Caspase-3 (1:1000,Cell Signaling Technology, 9662S), TBP (1:2000,Cell Signaling Technology, 44059S), FADD antibody was from Jianke Zhang's lab (1:2000, Thomas Jefferson University). Cell viability was determined by measuring ATP levels using the Cell Titer-Glo kit (Promega).

**Cell lines**. Mouse embryonic fibroblasts (MEFs) were cultured in Dulbecco's modified Eagle's medium (Hyclone, SH30243.LS) supplemented with 10% fetal bovine serum (Hyclone, SH30084.03),1% penicillin(100 IU/ml)/streptomycin (100 μg/ml) and 2 mM L-glutamine under 5% CO2 at 37 °C. Primary *Ripk1*$^{+/+}$ and *Ripk1*$^{K376R/K376R}$ MEFs were isolated from E11.5 littermate embryos. The head and visceral tissues were dissected and remaining bodies were incubated with 4 ml Tripsin/EDTA solution (Invitrogen, 25200072) per embryo at 37 °C for 1 h. After tripsinization, an equal amount of medium were mixed and pipetted up and down a few times. The primary MEFs were cultured in high-glucose DMEM containing 15% FBS. For immortalization, MEFs were transfected with SV40 small+large T antigen-expressing plasmid (Addgene, 22298) using Lipofectamine 2000 (Invitrogen, 11668019) according to manufacturer's instructions.

**Cell survival assay**. Cell survival was determined using the CellTiter-Glo Luminescent Cell Viability Assay kit (Promega, G7572) and the luminescence was recorded with a microplate luminometer (Thermo Scientific).

**Immunoblotting and immunoprecipitation**. Cells were harvested at different time points, washed with PBS and lysed with 1× SDS sample buffer containing 100 mM DTT. The cell lysates were cleared by centrifugation for 10 min at 12,000 × *g*, quantified by BCA kit (Thermo Scientific,23225), and then mixed with SDS sample buffer and boiled at 95 °C for 5 min. The samples were separated using SDS-PAGE, transferred to PVDF membrane (Millipore) with 110 v for 1.5 h. The proteins were detected by using a chemiluminescent substrate (Thermo Scientific). To immunoprecipitate RIP1, cells were lysed with lysis buffer (Tris-HCl 20 mM (pH 7.5), NaCl 150 mM, EDTA 1 mM, EGTA 1 mM, Triton X-100 1%, Glycerol 10%, Protease Inhibitor cocktail, Sodium pyrophosphate 2.5 mM, β-Glycerrophosphate 1 mM, NaVO$_4$ 1 mM, Leupeptin 1 μg/ml). Cell lysates were incubated for 3 h with 1 μg of RIP1 antibody (BD Biosciences, 610459). After mixing end over end for overnight (4 °C) with 35 μl of G-Agarose beads, the agarose was collected and washed three times with lysis buffer. Immunoprecipitates were denatured in SDS, subjected to SDS-PAGE, and immunoblotted. To immunoprecipitate Complex I, cells were harvest in NP-40 buffer (Tris-HCl 20 mM (pH 7.5), NaCl 120 mM, EDTA 1 mM, EGTA 1 mM, NP-40 0.5%, Glycerol 10%, Protease Inhibitor cocktail, Sodium pyrophosphate 2.5 mM, β-Glycerrophosphate 1 mM, NaVO$_4$ 1 mM, Leupeptin 1 μg/ml) after stimulating with 100 ng/ml TNF-α-FLAG(Enzo Life Science, ALX-522-009-C050). The cell lysates were then incubated with FLAG-tagged beads (Sigma, A2220) for 4 h. The beads were then collected and washed 3 times with NP-40 buffer. Immunoprecipitates were denatured in SDS, subjected to SDS-PAGE, and immunoblotted. The unprocessed scans of all blots are provided as Supplementary Fig. 6 in the Supplementary Information.

**MLKL oligomerization detection**. Cells were harvested after TSZ or TSZN treatment and lysed with 2× DTT-free sample buffer (Tris-Cl (PH 6.8) 125 mM, SDS 4%, Glycerol 20%, Bromophenol blue 0.02%) immediately. Total cell lysates were separated using SDS-PAGE, and transferred to PVDF membrane, and immunoblotting was performed with MLKL antibodies.

**Flow cytometry**. Lymphocytes were isolated from the thymus, spleen, and lymph nodes of mice. Antibodies against mouse CD3 (1:1000, eBioscience,11-0031-82), CD4 (1:1000, eBioscience,47-0042-82), CD8 (1:1000,Biolegend,100732), B220 (1:1000, eBioscience,12-0452-83), Gr-1 (1:1000, eBioscience, 12-5931-83), CD11b (1:1000, eBioscience, 17-0112-83) were used for flow cytometry analysis in this study. Single cell suspension of lymphocytes was stained on ice for half an hour with fluorescence-conjugated antibodies in the staining buffer (1× PBS, 3% BSA, 1 mM EDTA, 0.1%NaN3). After staining, cells were washed twice with PBS and immediately analyzed by flow cytometry (FACS Aria III, BD biosciences).

**RT-PCR**. Total RNA was extracted using Trizol reagent (Life Technologies), according to the manufacturer's instructions. After quantification, 1 μg total RNA was reverse transcribed to complementary DNA (Takara). Transcript levels of indicated genes were quantified by quantitative RT-PCR on an ABI 7500 real-time PCR instrument with SYBR Green. The sequences of primers are shown as follows. *Mcp-1*: 5′-AGGTGTCCCAAAGAAGCTGTA-3′ and 5′-ATGTCTGGACCCATTC CTTCT-3′; *A20*: 5′-CTCAGAACCAGAGATTCCATGAAG-3′ and 5′-CCTGTGTA GTTCGAGGCATGT-3′; *Icam-1*: 5′-CTTGGTAGAGGTGACTGA-3′ and 5′- GCT GAAAAGTTGTAGACTG-3′; *IκBα*: 5′-GATCCGCCAGGTGAAGGG-3′ and 5′-GCAATTTCTGGCTGGTTGG-3′; *Irf1*: 5′-CAGGGGAAAGAGAGAAAGTCC-3′ and 5′-CACACGGTGACAGTGCTGG-3′; *Il-1b*: 5′- CCCAACTGGTACATCAG CAC-3′ and 5′-TCTGCTCATTCACGAAAAGG-3′; *Ccl2*: 5′-TGAATGTGAAGTT GACCCGT-3′ and 5′-AAGGCATCACAGTCCGAGTC-3′; *Cxcl1*: 5′-CAAGAACA TCCAGAGCTTGAAGGT-3′ and 5′-GTGGCTATGACTTCGGTTTGG-3′; *Ifng*: 5′-ACAGCAAGGCGAAAAAGGAT-3′ and 5′-TGAGCTCATTGAATGCTTGG-3′;

**Statistical analysis**. Data presented in this article are representative results of at least three independent experiments. The statistical significance of data was evaluated by Student's *t* test and the statistical calculations were performed with GraphPad Prism software.

**Reporting summary**. Further information on research design is available in the Nature Research Reporting Summary linked to this article.

## Data availability
The authors declare that all data presented in this study are available within the figures and its Supplementary Information file. The source data underlying Figs. 3a, 4a, e, 5b, f, g, 6d, 7e, f and Supplementary Figs 1b, 2e, f are provided as a Source Data file. Other data that support the study are available from the corresponding author upon reasonable request.

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

## Acknowledgements
We thank Xiaodong Wang (National Institute of Biological Sciences, Beijing, China) for providing *Ripk3*$^{-/-}$ mice, Jianke Zhang (Thomas Jefferson University, Philadelphia, PA, USA) for providing *Fadd*$^{+/-}$ mice, Feng Shao (National Institute of Biological Sciences, Beijing, China) for providing *Tnfr1*$^{-/-}$ mice, and Qibin Leng (Institute Pasteur of Shanghai, Shanghai, China) for providing *Ifnar*$^{-/-}$ mice. We thank Junying Yuan for providing p-RIPK1(S166) antibody. We also thank Ronggui Hu (Shanghai Institute of Biochemistry and Cell Biology) and Hui Xiao (Institute Pasteur of Shanghai, Shanghai, China) for providing reagent and technical support. We thank Yu Sun (UCLA, Los

Angeles, California) for valuable suggestions and comments on this study. This work was supported by grants from the National Key Research and Development Program of China (2016YFC1304900, 2016YFA0500100) and the National Natural Science Foundation of China (31771537, 31571426).

## Author contributions

X.Z., Haiwei Z., and Haibing Z. conceived and designed the study; X.Z. and Haiwei Z. performed the experiments and analyzed data with assistance from C.X., X.L., M.L., W.P., and X.W.; B.Z. and Haikun W. analyzed data and provided technical support; D.L., Q.D., H.Y., and Hui W. provided essential reagents and intellectual input. X.Z., Haiwei Z., and Haibing Z. coordinated the project, interpreted results, and wrote the paper; Haibing Z. supervised the project.

## Additional information

**Competing interests:** The authors declare no competing interests.

