## [Peer Review File · Nature Communications]

Reviewers' comments:

Reviewer #1 (Remarks to the Author):

Zhang et al show that mutating the RIPK1 ubiquitination acceptor site Lys376 to Arg is lethal during mouse development. Lethality appears to be driven by TNFR1-triggered cell death because loss of TNFR1 or the combined loss of FADD/MLKL or FADD/RIPK3 allows the knock-in (KI) pups to be born. Consistent with what has already been shown in cell lines, the K376R mutation enhanced the sensitivity of primary MEFs to killing by TNF and impaired TNF-induced NF- κ B signaling.

More interesting is the finding that RIPK1 K376R KI, TNFR1 knockout (KO) mice succumb perinatally to RIPK3-dependent inflammation. As the authors point out, this could indicate that ubiquitination of RIPK1 is important for the suppression of RIPK3 activation in the perinatal period, similar to what has been shown for the RHIM of RIPK1.

Overall, I find this paper interesting, but there are several aspects that need improvement.

1) The authors suggest that apoptosis in the fetal liver is the main defect in the KI embryos (Fig. 1). However, the images provided in fig. 1C do not allow assessment of the yolk sac vasculature, which is often compromised when cell death pathways are perturbed during development (for example, as seen in the Casp8 KO, Ripk3 D161N KI, HOIP KO etc.) Indeed, images in fig. 4B would suggest that the yolk sac vasculature is compromised by RIPK1 K376R. Therefore, a better characterization of the KI yolk sac vasculature at E10-12 is warranted. In particular, is caspase-3 cleavage in KI endothelial cells enhanced when immunolabeling the yolk sac?

2) The WB in fig. 1E should include total RIPK1, not just p-RIPK1 S166.

3) Fig. 2A suggests that CYLD is more abundant in TNFR1 complex I of the RIPK1 K376R KI cells. CYLD is recruited via HOIP and SPATA2, so are these components also more abundant in complex I? This finding is unexpected because LUBAC is recruited through binding to polyubiquitin, but polyubiquitin on RIPK1 is shown to be reduced. What then is the underlying mechanism for this difference?

4) Fig. 2D shows that TNF-induced p-JNK is increased in K376R cells compared to wild-type cells. Is this difference reproducible? If yes, the authors should acknowledge this difference in the text.

5) Fig. 3C shows that RIPK1 K376R disappears from cell lysates after treatment with TSZ. The authors should indicate if this is due to RIPK1 K376R degradation or if RIPK1 K376R has moved into the insoluble fraction.

6) Fig. 4 should indicate by WB how much pRIPK1 S166 in RIPK1 K376R KI embryos was prevented by Nec1. In other words, was incomplete rescue of lethality due to incomplete inhibition of RIPK1 by Nec1, or might other mechanisms be involved?

7) In fig. 4F, the authors should plot the weights of males and females separately. Pooling sexes can skew the data if the sexes are not evenly represented, because males weigh more on average than females.

8) The legend to fig. 4G does not indicate the age of the mice analyzed.

9) The genetic evidence for RIPK3 driving lethality of RIPK1 KI, TNFR1 KO mice is compelling, but there is no evidence that RIPK3-dependent necroptosis is involved. The authors should WB or IHC stain tissues from the KI KO mice for pRIPK3 T231, S232 or pMLKL S345, which are hallmarks of necroptosis signaling, to clarify whether RIPK3 is mediating necroptosis versus other signaling

pathways.

10) Total leukocyte cellularity in thymus, spleen or lymph nodes should also be presented in Fig. S2D and S4D, because subset percentages in isolation do not provide a complete picture of what is happening in these tissues.

Reviewer #2 (Remarks to the Author):

Zhang et al. investigated the *in vivo* role of K376 RIPK1 by generating a K376R knockin mutant mouse line. Unexpectedly, the authors found that K377R mice died during embryogenesis. The authors showed that K376R MEFs demonstrated increased sensitivity to apoptosis and necroptosis when stimulated with TNF alone or a combination of SMAC and zVAD, reduced levels of RIPK1 ubiquitination in TNF-RSC, increased levels of activated RIPK1/RIPK3/MLKL and increased levels of complex II, which can be blocked by RIPK1 inhibitor Nec-1s. Consistently, the embryonic lethality of K376R mice was fully rescued by concomitant deletion of Fadd and Ripk3 or Mkl. The embryonic lethality of K376R mice was effectively prevented by treatment of RIPK1 kinase inhibitor and by deletion of Tnfr1 or one Ripk1K376R allele. However, Tnfr1^{-/-} Ripk1K376R/K376R mice displayed severe systemic inflammation and died around two weeks, which could be rescued by deletion of Ripk3. The authors conclude that Lys376-mediated ubiquitination of RIPK1 is important in suppressing RIPK1 kinase activity-dependent lethal pathways during embryonic development and RIPK3-dependent inflammation during postnatal development.

This is a thorough and well-designed study to investigate the role of K376 RIPK1 in regulating its ubiquitination and activation *in vivo*. The authors showed convincingly that blocking K376 ubiquitination of RIPK1 sensitizes to the activation of RIPK1, apoptosis and necroptosis in K376R mutant MEFs and leads to embryonic lethality. In fact, I believe that K376R mutation constitutes the first gain-of-function mutation in RIPK1 to promote its activation. While K376 was proposed to be important for mediating NF- κ B activation in TNF stimulated cells by previous studies (Ea et al. 2006), this manuscript demonstrated convincingly that ubiquitination at K376/377 RIPK1 is unexpectedly to be important for suppressing the activation of RIPK1 kinase instead. I recommend the acceptance of this manuscript.

Reviewer #3 (Remarks to the Author):

In their study, Zhang et al. aim at determining the *in vivo* function of RIPK1 ubiquitination on lysine K376. The authors describe that Ripk1K376R mice are embryonically lethal at E12.5. Given that Ripk1KO mice are born, the authors' observation is an important and exciting novel finding which may help clarify and dissect the pro-survival versus the pro-death functions of RIPK1.

Although embryonic lethality is rescued by ablation of TNFR1, Tnfr1KORipk1K376R mice still die postnatally. Notably, the inflammation observed in Tnfr1KORipk1K376R mice is due to aberrant necroptosis since deletion of RIPK3 was able to prevent this lethal inflammation. Ripk1K376R mice are completely rescued by ablation of FADD and MLKL or FADD and RIPK3, whilst these animals develop an *lpr*-like phenotype later on as expected.

Mechanistically, the authors demonstrate that Ripk1K376R cells are sensitized to TNF-induced death, display an impaired activation of NF- κ B and enhanced formation of complex II. The authors demonstrate that lack of RIPK1 ubiquitination induces RIPK1 phosphorylation and cell death induction. An interesting observation made by the authors is that Ripk1K376R^{-/-} mice were born although they develop fatal inflammation during adulthood. This goes to show that there is a tolerated threshold of RIPK1 phosphorylation that allows the mice to be born. However, later in life this induces cell death-

mediated inflammation.

Minor points:

1. The authors suggest that the RIPK1 376 mutation mainly perturbs K63 chains on RIPK1. However Figure S1A is a bit confusing. Why do the authors not see a decrease in overall ubiquitination in RIPK1 K376R Ub-WT condition (second lane)?
2. The images in Figure 4D are unclear and the authors should provide quantification.

Responses to Reviewers

We thank all three reviewers for their careful reading of our manuscript and their helpful and perceptive comments. In response to these comments, we have added new analysis and experimental data in the revised manuscript. Below we provide a point by point response to all of the reviewers' comments and concerns and believe these changes significantly strengthen the conclusions in the revised manuscript.

Reviewer #1:

1) The authors suggest that apoptosis in the fetal liver is the main defect in the KI embryos (Fig. 1). However, the images provided in fig. 1C do not allow assessment of the yolk sac vasculature, which is often compromised when cell death pathways are perturbed during development (for example, as seen in the Casp8 KO, Ripk3 D161N KI, HOIP KO etc.) Indeed, images in fig. 4B would suggest that the yolk sac vasculature is compromised by RIPK1 K376R. Therefore, a better characterization of the KI yolk sac vasculature at E10-12 is warranted. In particular, is caspase-3 cleavage in KI endothelial cells enhanced when immunolabeling the yolk sac?

We thank the reviewer for pointing out this issue. As suggested by the reviewer, we have checked the KI yolk sac vasculature at E12.5 using immunostaining of VE-cadherin (a marker of vascular endothelial cells) and cleaved caspase-3 (shown below). The results showed that the KI yolk sacs displayed vascular abnormalities with remarkably increased levels of cleaved caspase-3 which indicated that the endothelial cells underwent excessive apoptosis in KI yolk sacs. This suggests that K376R of RIPK1 has pronounced impacts on not only embryonic tissues but also yolk sacs. These data have now been included into the new Fig. 1e. Thus, we have addressed the reviewer's concern.

2) The WB in fig. 1E should include total RIPK1, not just p-RIPK1 S166.

We thank the reviewer for pointing out this issue. To address the reviewer's concern, we have examined total RIPK1 in the body tissues of wt and mutant embryos. We found that RIPK1 phosphorylation (S166) was consistently induced in KI embryonic tissues and total RIPK1 aggregated together with RIPK3 and transferred from soluble fraction to insoluble fraction, which is a hallmark of necroptosis (shown below). These data indicate that RIPK1 K376R causes not only apoptosis but also necroptosis during embryogenesis. These data and relevant description have been included into new Fig. 1f. Thus, we have addressed the reviewer's concern.

3) Fig. 2A suggests that CYLD is more abundant in TNFR1 complex I of the RIPK1 K376R KI cells. CYLD is recruited via HOIP and SPATA2, so are these components also more abundant in complex I? This finding is unexpected because LUBAC is recruited through binding to polyubiquitin, but polyubiquitin on RIPK1 is shown to be reduced. What then is the underlying mechanism for this difference?

We thank the reviewer for this insightful comment. To address the reviewer's concern, we repeated this experiment several times and confirmed that the levels of CYLD in wt and KI cells were comparable (as shown below), suggesting that the defect in RIPK1 K376-mediated ubiquitination had no effect on CYLD recruitment into complex I. We have now included this new Figure.2a in the revised version.

4) Fig. 2D shows that TNF-induced p-JNK is increased in K376R cells compared to wild-type cells. Is this difference reproducible? If yes, the authors should acknowledge this difference in the text.

We thank the reviewer for pointing out this issue. Per the reviewer's suggestion, we have repeated this experiment several times and confirmed that TNF-induced p-JNK was comparable between wt cells and KI cells (as shown below). We have included a representative result as new Figure.2d in the revised version.

5) Fig. 3C shows that RIPK1 K376R disappears from cell lysates after treatment with TSZ. The authors should indicate if this is due to RIPK1 K376R degradation or if RIPK1 K376R has moved into the insoluble fraction.

We thank the reviewer for pointing out this issue. Per the reviewer's suggestions, we separated the soluble fraction and insoluble fraction after TSZ treatment and found that RIPK1 and RIPK3 were clearly detected in the insoluble fraction in KI cells which subsequently led to the phosphorylation and oligomerization of MLKL to execute necroptosis (as shown below). This process occurred much earlier in KI cells than in wt controls, suggesting that RIPK1 K376R promotes necroptosis in response to TSZ treatment. We have added this data in the revised Figure.3c. Thus, we have addressed the reviewer's concern.

6) Fig. 4 should indicate by WB how much pRIPK1 S166 in RIPK1 K376R KI embryos was prevented by Nec1. In other words, was incomplete rescue of lethality due to incomplete inhibition of RIPK1 by Nec1, or might other mechanisms be involved?

We thank the reviewer for bringing up this important point. To address reviewer's concern, we have performed additional analysis with tissue of embryos by western blotting. We found that the level of p-RIPK1(S166) in KI embryos was significantly reduced but not completely blocked by Nec-1s treatment (shown below). Therefore, we concluded that the incomplete rescue may due to incomplete inhibition of RIPK1 kinase activity by Nec-1s and included this new data in Fig.4f. However, a definitive answer will require extensive genetic studies through the generation of RIPK1^{D138N}RIPK1^{K376R} double mutant mice, which is beyond the scope of the present study.

7) In fig. 4F, the authors should plot the weights of males and females separately. Pooling sexes can skew the data if the sexes are not evenly represented, because males weigh more on average than females.

We thank the reviewer for pointing out this point. Per reviewer's request, we plotted the body weights of males and females respectively and figured out that RIPK1 K376R caused phenotypes were evenly represented in both males and females (as shown below). The new data are presented in new Figure.5f.

8) The legend to fig. 4G does not indicate the age of the mice analyzed.

We understand the reviewer for this omission in Fig.5G and we have now added the age of the mice in the revised figure legend.

9) The genetic evidence for RIPK3 driving lethality of RIPK1 KI, TNFR1 KO mice is compelling, but there is no evidence that RIPK3-dependent necroptosis is involved. The authors should WB or IHC stain tissues from the KI KO mice for pRIPK3 T231, S232 or pMLKL S345, which are hallmarks of necroptosis signaling, to clarify whether RIPK3 is mediating necroptosis versus other signaling pathways.

We thank the reviewer for bringing up this important point. In response, we have performed the following experiments, we isolated BMDMs from *Tnfr1*^{-/-}*Ripk1*^{K376R/K376R} mice and their littermate controls and stimulated them with LPS or Poly(I:C) plus zVAD to evaluate the necroptosis signaling in these cells. The result showed that the *Tnfr1*^{-/-}*Ripk1*^{K376R/K376R} cells were much more vulnerable compared to their controls and the phosphorylations of RIPK1 and RIPK3(pRIPK3 S232) were more significantly induced in these cells (as shown below). The result suggests that RIPK3-mediated necroptosis plays an important role under RIPK1 KI, TNFR1 KO circumstance. We have included these data in new Fig. S2f, S2g to illustrate the role of RIPK3 in *Tnfr1*^{-/-}*Ripk1*^{K376R/K376R} mice. Thus, we have addressed the reviewer's concern.

10) Total leukocyte cellularity in thymus, spleen or lymph nodes should also be presented in Fig. S2D and S4D, because subset percentages in isolation do not provide a complete picture of what is happening in these tissues.

We thank the reviewer for pointing out this issue. Per the reviewer's recommendation, we have presented total cell numbers of mice in Fig. S2D and S4D and the data have now shown in new Fig.S2e and Fig. 7e. Thus, we have addressed the reviewer's concern.

Reviewer #2:

Zhang et al. investigated the *in vivo* role of K376 RIPK1 by generating a K376R knockin mutant mouse line. Unexpectedly, the authors found that K377R mice died during embryogenesis. The authors showed that K376R MEFs demonstrated increased sensitivity to apoptosis and necroptosis when stimulated with TNF alone or a combination of SMAC and zVAD, reduced levels of RIPK1 ubiquitination in TNF-RSC, increased levels of activated RIPK1/RIPK3/MLKL and increased levels of complex II, which can be blocked by RIPK1 inhibitor Nec-1s. Consistently, the embryonic lethality of K376R mice was fully rescued by concomitant deletion of *Fadd* and *Ripk3* or *Mlkl*. The embryonic lethality of K376R mice was effectively prevented by treatment of RIPK1 kinase inhibitor and by deletion of *Tnfr1* or one *Ripk1*^{K376R} allele. However, *Tnfr1*^{-/-} *Ripk1*^{K376R/K376R} mice displayed severe systemic inflammation and died around two weeks, which could be rescued by deletion of *Ripk3*. The authors conclude that Lys376-mediated ubiquitination of RIPK1 is important in suppressing RIPK1 kinase activity-dependent lethal pathways during embryonic development and RIPK3-dependent inflammation during postnatal development.

This is a thorough and well-designed study to investigate the role of K376 RIPK1 in regulating its ubiquitination and activation *in vivo*. The authors showed convincingly that blocking K376 ubiquitination of RIPK1 sensitizes to the activation of RIPK1, apoptosis and necroptosis in K376R mutant MEFs and leads to embryonic lethality. In fact, I believe that K376R mutation constitutes the first gain-of-function mutation in RIPK1 to promote its activation. While K376 was proposed to be important for mediating NF- κ B activation in TNF stimulated cells by previous studies (Ea et al. 2006), this manuscript demonstrated convincingly that ubiquitination at K376/377 RIPK1 is

unexpectedly to be important for suppressing the activation of RIPK1 kinase instead. I recommend the acceptance of this manuscript.

We thank the reviewer for these comments.

Reviewer #3:

1. The authors suggest that the RIPK1 376 mutation mainly perturbs K63 chains on RIPK1. However Figure S1A is a bit confusing. Why do the authors not see a decrease in overall ubiquitination in RIPK1 K376R Ub-WT condition (second lane)?

We thank the reviewer for pointing out this point. To address the reviewer's concern, we have performed the experiment and confirmed that K376R mainly perturbs K63 chains on RIPK1. The result shown below represents three repeats which show that the total ubiquitinations and K63-linked ubiquitinations are decreased while K48-linked ubiquitinations remain unchanged. We have included a representative results as new Fig.S1a in the revised version.

2. The images in Figure 4D are unclear and the authors should provide quantification.

We thank the reviewer for pointing out this issue. Per the reviewer's request, we have now quantified the images in Fig.4D and present in new Figure. 4e. Thus, we have addressed the reviewer's concern.

REVIEWERS' COMMENTS:

Reviewer #1 (Remarks to the Author):

A few things still need addressing:

1. The figure legend to 1e does not indicate the age of the embryos/yolk sacs analysed. Nor how many of each genotype were stained.

2. Treating BMDMs with a necroptotic stimulus in Supplementary Fig. 2f, g provides no direct insight into the involvement of necroptosis in the TNFR1 KO RIPK1 KI pups. There are many studies now that indicate RIPK3 does more than just trigger MLKL-dependent necroptosis to promote inflammation. Proof would be that MLKL deficiency provides the same protection as RIPK3 loss. I realize this is beyond the scope of the present study BUT they can readily WB or stain tissues for p-RIPK3 or p-MLKL, the hallmarks of necroptosis signaling. Whether the data is positive or negative, it is informative and should be included.

Reviewer #3 (Remarks to the Author):

In the revised version of their manuscript, Zhang et al. have addressed all of my major concerns raised with regarding the pervious version of the manuscript. The results presented in this study and sound and advance our understanding of the molecular mechanisms that determine the switch between the pro-survival and the pro-death functions of RIPK1.

REVIEWERS' COMMENTS:

Reviewer #1 (Remarks to the Author):

A few things still need addressing:

1. The figure legend to 1e does not indicate the age of the embryos/yolk sacs analyzed. Nor how many of each genotype were stained.

We thank the reviewer for pointing out this omission. We have added the description to indicate the age (E12.5) and number of the embryos and yolk sacs analyzed (n=3/genotype) in the revised figure legend of Figure 1e.

2. Treating BMDMs with a necroptotic stimulus in Supplementary Fig. 2f, g provides no direct insight into the involvement of necroptosis in the TNFR1 KO RIPK1 KI pups. There are many studies now that indicate RIPK3 does more than just trigger MLKL-dependent necroptosis to promote inflammation. Proof would be that MLKL deficiency provides the same protection as RIPK3 loss. I realize this is beyond the scope of the present study BUT they can readily WB or stain tissues for p-RIPK3 or p-MLKL, the hallmarks of necroptosis signaling. Whether the data is positive or negative, it is informative and should be included.

We thank the reviewer for pointing out this issue. Indeed, we had examined the p-RIPK3 and p-MLKL in multiple tissues of TNFR1 KO RIPK1 KI mice. However, the antibodies cannot detect the phosphorylated RIPK3 or MLKL of tissues specifically. Thus, we agree that this is an important issue to be addressed in the future. Therefore we have added interpretations regarding this issue in the revised discussion section.

Reviewer #3 (Remarks to the Author):

In the revised version of their manuscript, Zhang et al. have addressed all of my major concerns raised with regarding the previous version of the manuscript. The results presented in this study and sound and advance our understanding of the molecular mechanisms that determine the switch between the pro-survival and the pro-death functions of RIPK1.

We thank the reviewer for these comments.